# What determines the sign of the evapotranspiration response to afforestation in the European summer?

Marcus Breil[1], Edouard L. Davin[2], Diana Rechid[3]

[1]Institute for Meteorology and Climate Research, Karlsruhe Institute of Technology, Karlsruhe, Germany
[2]Department of Environmental Systems Science, ETH Zurich, Zurich, Switzerland
[3]Climate Service Center Germany, Helmholtz-Zentrum Geesthacht, Hamburg, Germany

*Correspondence to*: Marcus Breil (marcus.breil@kit.edu)

**Abstract.** Uncertainties in the evapotranspiration response to afforestation constitute a major source of disagreement between model-based studies of the potential climate benefits of forests. Forests typically have higher evapotranspiration rates than grassland in the tropics, but whether this is also the case in the mid-latitudes is still debated. To explore this question and the
underlying physical processes behind these varying evapotranspiration rates of forests and grasslands in more detail, a regional model study with idealized afforestation scenarios was performed for Europe. In the first experiment Europe was maximally forested and in the second one, all forests were turned into grassland.

The results of this modelling study exhibit the same contradicting evapotranspiration characteristics of forests and grasslands as documented in observational studies. But by means of an additional sensitivity simulation, in which the surface roughness
of forest was reduced to grassland, the mechanisms behind these varying evapotranspiration rates could be revealed. Due to the higher surface roughness of a forest, solar radiation is more efficiently transformed into turbulent sensible heat fluxes, leading to lower surface temperatures (top of vegetation) than in grassland. The saturation deficit between the vegetation and the atmosphere, which depends on the surface temperature, is consequently reduced over forests. This reduced saturation deficit counteracts the transpiration facilitating characteristics of a forest (deeper roots, a higher LAI and lower albedo values
than grassland). If the impact of the reduced saturation deficit exceeds the effects of the transpiration facilitating characteristics of a forest, evapotranspiration is reduced compared to grassland. If not, evapotranspiration rates of forests are higher. The interplay of these two counteracting factors depends on the latitude and the prevailing forest type in a region.

## 1 Introduction

Afforestation is frequently discussed as a potential strategy to mitigate the effects of human-induced climate change (e.g. Sonntag et al., 2016; Harper et al., 2018; Roe et al., 2019; Davin et al., 2020). One benefit of afforestation is that forests are generally able to take up more $CO_2$ than grasslands (IPCC, 2019). Another advantage is that forests can have a cooling effect on the land surface due to increased evapotranspiration rates compared to grasslands (e.g. Bonan, 2008; Bright et al., 2017; Duveiller et al., 2018). According to our present knowledge about the biogeophysical effects of forests and grasslands, this
increased forest evapotranspiration is caused by deeper roots (Schenk and Jackson, 2003) and a higher Leaf Area Index (LAI, e.g. Henderson-Sellers, 1993) than in grassland, whose influence can be attenuated by a reduced photosynthetic activity of

forests and an associated stomata closure (Leuzinger et al., 2005). The evaporative cooling effect is particularly pronounced in the tropics (Von Randow et al., 2004) but is unclear at mid-latitudes (Bonan, 2008). While several observation-based studies show higher evapotranspiration rates of forests at mid-latitudes (e.g. Zhang et al., 2001; Li et al., 2015; Chen et al., 2018; Duveiller et al., 2018), some studies exhibit an opposite behavior of forests with reduced evapotranspiration rates compared to grasslands (e.g. Wicke and Bernhofer, 1996; Teuling et al., 2010; Williams et al., 2012). The actual evapotranspiration rates of forests and grasslands are therefore subject of controversial discussions within the scientific community (e.g. Teuling, 2018).

An adequate methodology to improve the understanding about this contradicting evapotranspiration responses, is the application of model simulations, in which factorial experiments are performed in order to disentangle the role of different processes. But also within executed model intercomparison studies, a number of models simulate increased evapotranspiration, some models simulate decreased evapotranspiration in forests during summer (de Noblet-Ducoudré et al. 2012; Lejeune et al., 2017; Davin et al., 2020). The mechanisms behind the diverging evapotranspirative behavior of forests and grasslands in the mid-latitudes are consequently still an unsolved issue. Thus, to be able to correctly assess the suitability of afforestation as an effective mitigation strategy in the mid-latitudes, the understanding of the biogeophysical processes in forests and grasslands need to be improved. Only if the evapotranspirative behavior of forests and grasslands can be properly explained, the impact of these land use types on the near surface climate conditions can be evaluated.

In this study, therefore, the question how afforestation can lead in some parts of the mid-latitudes to increased evapotranspiration rates in summer and in some regions to a reduction, will be further explored. For this, idealized and extensive afforestation scenarios are applied in regional climate simulations for Europe. This approach allows an isolated view on the biogeophysical processes in forest and grasslands on a large scale, which is not provided by selective point observations. The theoretical background of the transpiration flux calculation and the simulation setup of the afforestation experiments is provided in section 2. Based on the presented simulation results in section 3, a mechanism explaining the varying evapotranspiration rates of forest and grasslands is discussed in section 4.

## 2 Method

To investigate the processes determining the sign of the evapotranspiration response to afforestation in the mid-latitudes, simulations with the Regional Climate Model COSMO-CLM (Rockel et al., 2008), coupled to the Land Surface Model (LSM) VEG3D (Breil and Schädler, 2017) are performed for Europe. Since afforestation is primarily affecting the transpiration characteristics of a land surface, it is assumed that changes in total evapotranspiration in summer are mainly caused by changes in the transpiration rates as indicated e.g. by Meier et al., (2018). The focus of the paper will therefore be on the impact of afforestation on transpiration changes and evapotranspiration responses are tried to be explained by changes in the transpiration characteristics. According to this, in a first step, the theoretical background of transpiration is presented and its implementation in the LSM VEG3D is discussed in detail. Subsequently, the setup of the performed simulations is described.

### 2.1 Theoretical background

Transpiration can be described as a water flux from a vegetated land surface into the atmosphere. This flux is determined by two factors: (1) the saturation deficit between the vegetation and the atmosphere $q_s(T_{scf})$ - $q_a$, and (2) a transfer coefficient $c$:

$$Q = p * c\left(q_s(T_{scf}) - q_a\right)$$ Eq. (1)

$q_s(T_{scf})$ depends on the surface temperature $T_{scf}$ and is derived from the Magnus-Equation. The surface temperature is in this case the temperature at the top of the vegetation. $p$ is the air density. In state of the art LSMs, the transfer coefficient $c$ is generally regarded as a resistance that has to be overcome by the transpiration flux (e.g. Niu et al. (2011); Oleson et al. (2013)).

In VEG3D, the LSM applied in this study, this drag coefficient is described through two resistances in series (Deardorff, 1978 and Taconet et al., 1986), an atmospheric resistance $r_a$ and a canopy resistance $r_c$:

$$c = \frac{frac_{dry}}{r_c + r_a}$$ Eq. (2)

$frac_{dry}$ represents the fraction of dry leaf surface.

In $r_a$, the turbulent atmospheric conditions for the transfer of water vapor are included, which are calculated by means of an empirical parameter $C_{leaf}$ and the friction velocity $u_*$ :

$$r_a = \frac{C_{leaf}}{u_*}$$ Eq. (3)


$C_{leaf}$ describes an empirical interrelation between the turbulent exchange and the Leaf Area Index ($LAI$) (Taconet et al., 1986), in relation to the leaf geometry, represented by the plant specific parameter $c_{veg}$ (Goudriaan, 1977):

$$C_{leaf} = \frac{1 + 0.5 * LAI}{0.04 * LAI * c_{veg}}$$ Eq. (4)


$u_*$ is classically derived from the Monin-Obukhov Similarity Theory (Monin and Obukhov, 1954) and as such mainly dependent on $z_0$:

$$u_* = \frac{k(v_{z_a} - v_{z_0})}{\ln\left(\frac{z_a}{z_0}\right) + \Psi\left(\frac{z_a}{L_*}\right) + \Psi\left(\frac{z_0}{L_*}\right)}$$ Eq. (5)


where $z_a$ is the height of the lowest atmospheric model level and $z_0$ is the roughness length. $v_{z_a}$ and $v_{z_0}$ are consequently the wind velocities at the respective heights. $k$ is the Karman-constant. $L_*$ is the Monin-Obukhov length and $\Psi$ is a stability-function according to Businger et al., (1971), establishing empirical relationships in turbulent motion, which depend on the atmospheric stratification. According to Eq. (4) and Eq. (5), $r_a$ is consequently affected by three vegetation parameters, namely

a plant specific parameter $c_{veg}$ (a), the surface roughness $z_0$ (b) and the $LAI$ (c). But out of these three parameters, the influence of the surface roughness (b) on $r_a$ and thus, on the transfer coefficient $c$, is clearly dominating (Goudriaan 1977).

In $r_c$, the plant physiological processes of transpiration are considered. Soil water is thereby extracted by the roots and transported into the leaves. There, the water is released through the stomata into the atmosphere. Plants are regulating this water flux by the closure of the stomata. In the case of high solar radiation, for instance, stomata can be opened to increase

the evaporative cooling. On the other hand, in the case of limited water availability, the stomata can be closed and transpiration is reduced. These different canopy functions are described by $r_c$ (Deardorff, 1978 and Taconet et al., 1986):

$$r_c = r_{min} \frac{1+0.5*LAI}{LAI} \left( \frac{S_{max}}{S+0.03*S_{max}} + \left(\frac{w_{wilt}}{w_{root}}\right)^2 \right) \qquad \text{Eq. (6)}$$

$r_c$ depends on the net short-wave radiation, whereby $S$ is the actual net short-wave radiation and $S_{max}$ constitutes a seasonally varying maximum short-wave radiation. Vegetation affects these components by the albedo parameter (d). In VEG3D, a bulk surface albedo with prescribed parameter values is used, depending on the vegetation type. Additionally, $r_c$ depends on the soil water availability, which is described by the relation of the wilting point $w_{wilt}$ to the soil water content within the rooted soil $w_{root}$. Vegetation affects the soil water content by the root depth parameter (e). Furthermore, $r_c$ is controlled by the *LAI* (c)

and a plant specific stomata coefficient $r_{min}$ (f), representing plant specific stomatal resistance characteristics (Deardorff, 1978).

Thus, in VEG3D transpiration depends on six different vegetation parameters (a-f), besides the humidity gradient (1) (Table 1). The values of these six vegetation parameters in VEG3D are in line with the parameter values used in other state of the art LSMs (Breil et at., 2020). In a forest, these vegetation characteristics are different to grassland:

•   Trees have generally larger leaves than grass. The leaf geometry parameter $c_{veg}$ is therefore higher for forests than for grasslands (Taconet et al., 1986). Thus, $r_a$ is reduced and transpiration is facilitated.

•   The surface roughness of a forest is higher than of grassland (Garratt, 1993; Henderson-Sellers, 1993). The turbulent mixing is consequently increased, what in turn reduces $r_a$ and facilitates transpiration.

•   The *LAI* for forest is higher than for grassland (e.g. Henderson-Sellers, 1993). With a high *LAI*, more water can be

transpired. The canopy resistance $r_c$ of forest is therefore reduced. Furthermore, a high *LAI* increases interception, what additionally increases evapotranspiration.

•   A forest is characterized by lower albedo values than grassland (Garratt, 1993; Henderson-Sellers, 1993). Thus, the net short-wave radiation $S$ is increased. This leads particularly in summer to a reduced canopy resistance $r_c$, what facilitates transpiration.

•   The roots in a forest reach deeper than in grassland (Schenk and Jackson, 2003). During dry summer conditions, therefore, the available amount of water for transpiration is increased in a forest. The water stress for the trees is consequently low, leading again to a reduced $r_c$.

•   Values of $r_{min}$ for forest and grassland vary in literature, but are on a similar level in VEG3D as stated by Garratt, (1993). In the presented study, a lower $r_{min}$ for forest is used than for grassland, leading to lower $r_c$ values under the

same boundary conditions.

Thus, each of the six factors (a-f), which affect the transfer coefficient $c$ (Eq. 2) in the transpiration flux calculation (Eq. 1) in VEG3D, is reduced in forest compared to grassland and thus, facilitates transpiration during summer. According to Eq. (1), a reduced transpiration in a forest must consequently be connected to a reduced saturation deficit between the vegetation and

the atmosphere. In the following, therefore, the impact of this saturation deficit on the transpiration fluxes of forests and grasslands and its relations to the vegetation parameters (a-f) is investigated. For this, an idealized model study is conducted, to explore the reasons for the uncertain effects of afforestation in European summer.

**Table 1:** The impact of the different influencing factors on transpiration of forests in comparison to grasslands

| Parameter | Impact on transpiration |
|---|---|
| leaf geometry | facilitates transpiration |
| surface roughness | facilitates transpiration |
| LAI | facilitates transpiration |
| albedo | facilitates transpiration |
| root depth | facilitates transpiration |
| stomatal resistance | facilitates transpiration |
| saturation deficit | attenuates transpiration |


## 2.2 Simulation Setup

As described in the previous section, transpiration depends on two factors, (1) the saturation deficit between the surface and the atmosphere and (2) the transfer coefficient $c$. (2) can thereby be described by two resistances $r_a$ and $r_c$, which are controlled by six vegetation parameters (a-f). Now, the impact of all these components on the transpiration flux of forests and grasslands

is investigated, by performing idealized afforestation simulations with a regional climate model.

For this, two extreme land use change scenarios for Europe are simulated. In the first experiment, Europe is completely covered with forest, where trees can realistically grow (FOREST), in the second experiment all forest is turned into grassland (GRASS). By using this approach, the differences in transpiration between forests and grasslands can be isolated and analyzed on a large scale, which is not given in observation studies. In this way, the mechanisms leading to the different transpiration

responses to afforestation in the European summer can be explored in detail.

In FOREST, two different forest types are used (coniferous and deciduous), in GRASS only one grassland class is applied. The spatial distribution of the two different forest types in FOREST is illustrated in Figure 1. Coniferous and deciduous forest, as well as grassland, have different vegetation characteristics, leading to different transpiration rates, as already described in section 2.1. The used vegetation parameters for each land use class are summarized in Table 2. The study is embedded in the

LUCAS initiative (Rechid et at., 2017). The model domain is the Coordinated Downscaling Experiment-European Domain (EURO-CORDEX; Jacob et al., 2014), in a horizontal resolution of 0.44° (~50km). The simulations were driven by ERA-Interim reanalyses (Dee et al., 2011) at the lateral boundaries and at the lower boundary over sea. The simulation period is 1986-2015. A spin-up of six years was performed before 1986.

To be able to better distinguish between the effects of $r_a$ and $r_c$ on the respective transpiration fluxes, an additional sensitivity

run with the FOREST setup is performed (ROUGH). In this simulation the surface roughness of forest is replaced by the surface roughness of grassland. All the other vegetation parameters of forest, like albedo or *LAI*, remained unchanged. Since the surface roughness affects only $r_a$ and not $r_c$, this sensitivity simulation gives the opportunity to draw conclusions about the impact of both resistances on the transpiration fluxes.

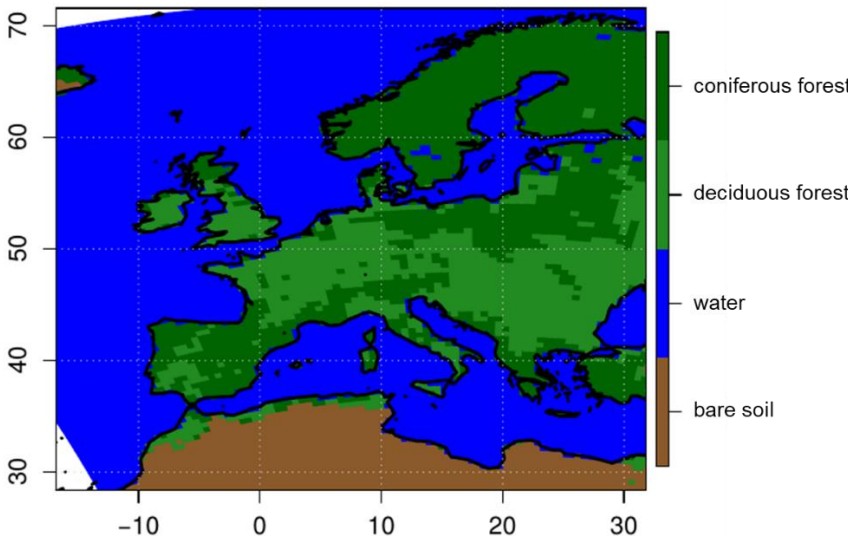

Figure 1: Spatial distribution of the land use classes used in the FOREST experiment.

Table 2: Vegetation parameters of the different land use classes in summer.

|  | Albedo | *LAI* | $r_{min}$ | root depth (density < 2%) | $z_0$ | $c_{veg}$ |
|---|---|---|---|---|---|---|
| Coniferous forest | 0.11 | 9 | 120 | 1.0 m | 1.0 m | 1.75 |
| Deciduous forest | 0.15 | 8 | 120 | 2.0 m | 0.8 m | 2.1 |
| Grassland | 0.2 | 4 | 150 | 0.5 m | 0.03 m | 1.2 |

## 3 Results

### 3.1 Evapotranspiration

In Southern and Central Europe, evapotranspiration is reduced in the FOREST run compared to the GRASS simulation (Figure 2a). The evapotranspiration reduction in FOREST is in this context particularly strong in Southern Europe. But in Northern Europe the opposite is the case and evapotranspiration is increased in FOREST. In Central Europe, regions with reduced evapotranspiration rates in FOREST coincide with regions covered by deciduous forest (Figure 1). This indicates that differences in evapotranspiration rates between forests and grassland are affected by the prevailing forest type in a region. Thus, the different vegetation characteristics (a-f) of deciduous and coniferous forest, must have an impact on the intensity of the evapotranspiration response to afforestation. But since both forest types have lower resistance values (higher *c* values) than grasslands, both forest types should also stronger promote transpiration than grasslands, which seems to be in contradiction to the reduced evapotranspiration rates of deciduous forests in Central Europe. Therefore, the resistance values of the different forest types cannot solely explain the opposing transpiration signals.

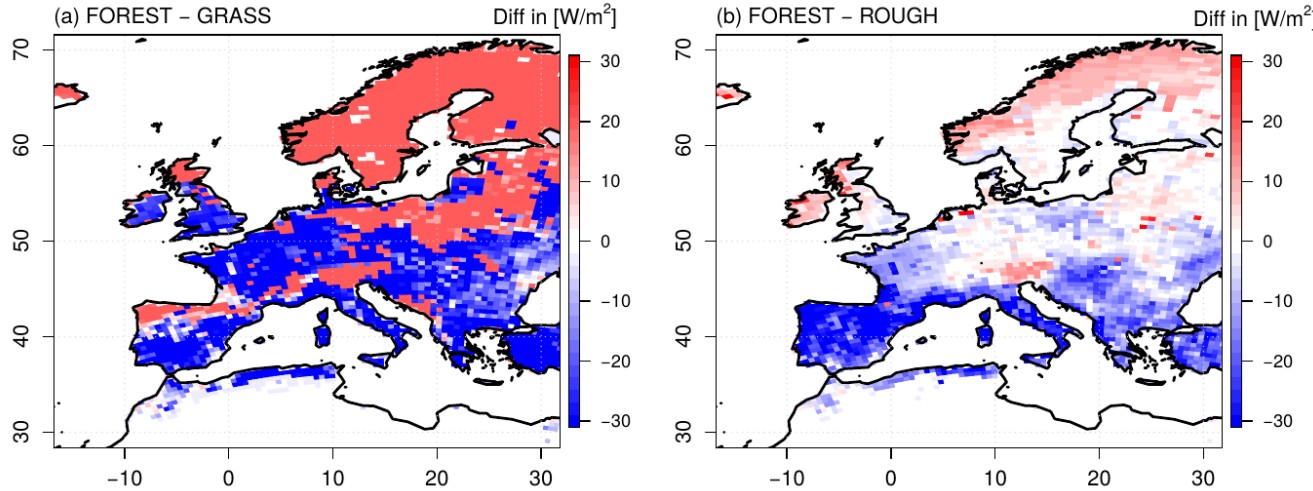

**Figure 2:** Differences in mean seasonal latent heat fluxes in summer between the FOREST and the GRASS experiment (a), and the FOREST and the ROUGH experiment (b), over the simulation period 1986-2015.

In general, differences in evapotranspiration rates are frequently connected to differences in the soil water contents and thus, differences in the amount of available water for evapotranspiration. But due to their deeper roots, forests have access to a larger amount of available soil water than grasslands (Figure 3a), so that the drought stress in summer is lower in the FOREST simulation than in the GRASS run. The reduced evapotranspiration rates in Central and Southern Europe in FOREST can consequently not be caused by lower soil water contents.

Furthermore, by means of differences in the soil water content, the contribution of transpiration and soil evaporation to total evapotranspiration can be indirectly assessed. Figure 3b-d show the differences in soil water contents between the FOREST simulation and the GRASS run for different soil depths. Differences in the upper 5 cm of the soil (Figure 3b) are used as an indicator for differences in the soil evaporation, since this process is executed through the soil surface (although soil evaporation can also be affected by soil depths deeper than 5 cm). In a depth of 15 cm (Figure 3c) the maximum root density of grassland is located, in 75 cm depth (Figure 3d) the maximum of forest. Thus, differences in these soil depths refer to the contribution of transpiration to total evaporatranspiration in each simulation. Just slight differences occur between the FOREST and the GRASS simulation for the upper soil (Figure 3b). This is because the upper soil layers are in both simulations almost completely dry in summer. The contribution of soil evaporation to total evapotranspiration is therefore low in both simulations. This confirms the proposed assumption at the beginning of the study (section 2) that changes in total evapotranspiration in summer are mainly associated to transpiration. In a depth of 15 cm, almost all over Europe the soil is drier in the GRASS simulation (Figure 3 c), since grassland extracts water for transpiration mainly from this depth. The same applies to forest in 75 cm depth (Figure 3d). But since forest is, in contrast to grassland, able to extract water from these deeper soil layers, the available soil water amount for transpiration in summer is higher in FOREST than in GRASS (Figure 3a).

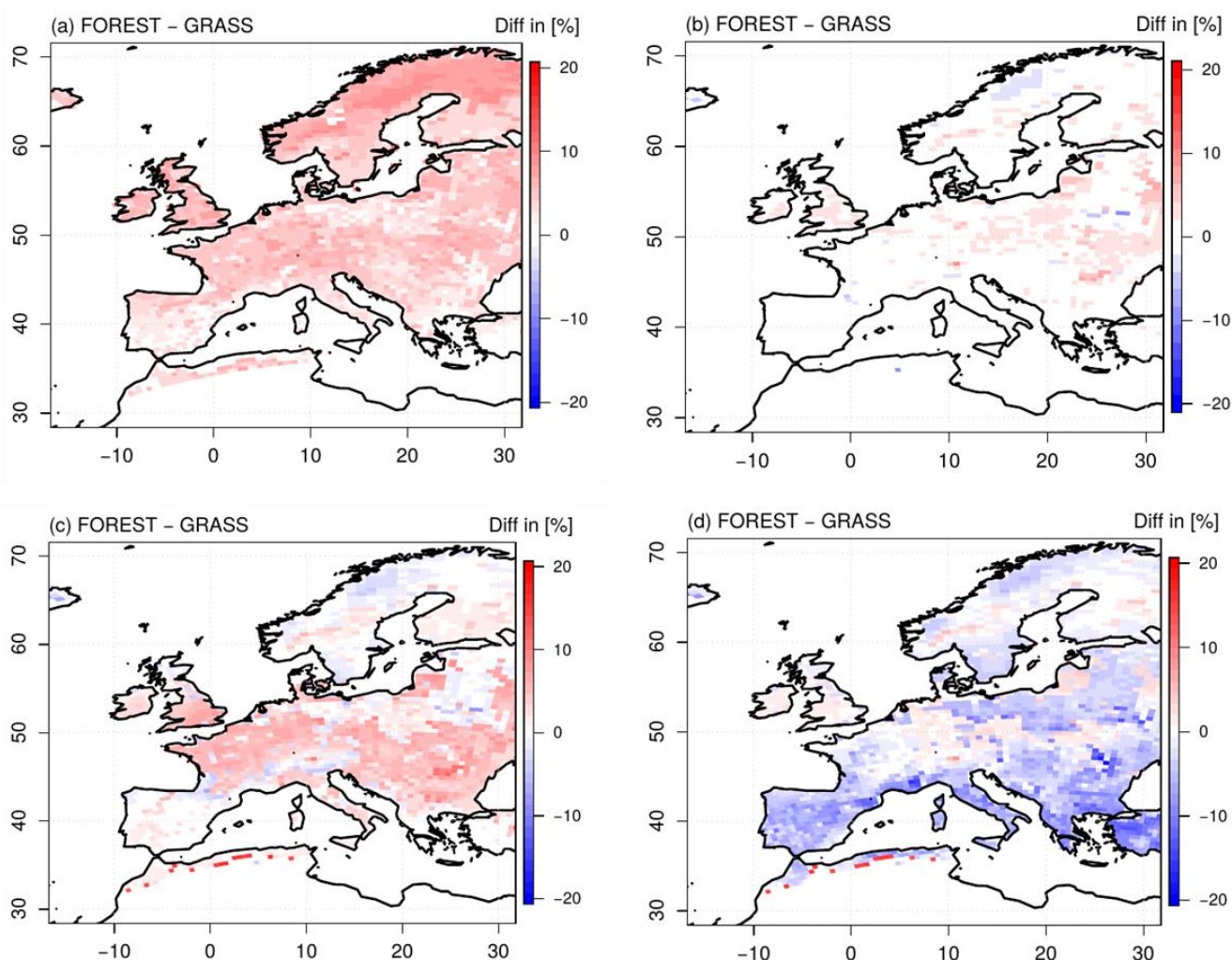

**Figure 3:** Differences between the FOREST and the GRASS experiment in summer for the available soil water amount for evapotranspiration (soil water content – residual soil water content) within the rooted soil column (a), and the upper soil layers (until 5 cm depth) (b), a soil depth of 15 cm (c), a soil depth of 75 cm (d), over the simulation period 1986-2015.

The ROUGH sensitivity simulation, with its reduced surface roughness, provides the opportunity to additionally investigate the impact of the resistance part $r_a$ on the transpiration flux more precisely (Figure 2b). In general, a reduced surface roughness reduces turbulent mixing, which is manifested in an increased $r_a$. According to Eq. (2), this reduces the transfer coefficient $c$ and transpiration is impeded. This should consequently lead all over Europe to reduced transpiration rates in ROUGH. But this is only the case in Northern Europe. In Southern Europe and large parts of Central Europe evapotranspiration is even increased compared to FOREST. Thus, the ROUGH simulation exhibits astonishingly comparable evapotranspiration patterns to the GRASS run and does not anymore behave like a forest simulation. Since an increase in $r_a$ should have an opposite effect, its impact on the transpiration flux signal must be negligible, at least in Southern and Central Europe. But the generally strong effects of the surface roughness change on evapotranspiration indicates that surface roughness is playing a major role for evapotranspiration beyond its impact on $r_a$.

### 3.2 Saturation deficit

According to Eq. (1), the saturation deficit between the vegetation and the atmosphere is the driving force of transpiration, which is regulated by the transfer coefficient $c$. In the FOREST simulation, this saturation deficit is all over Europe reduced compared to the GRASS simulation (Figure 4a). Thus, all over Europe, the transpiration facilitating vegetation characteristics of a forest are facing a reduced driving force of transpiration.

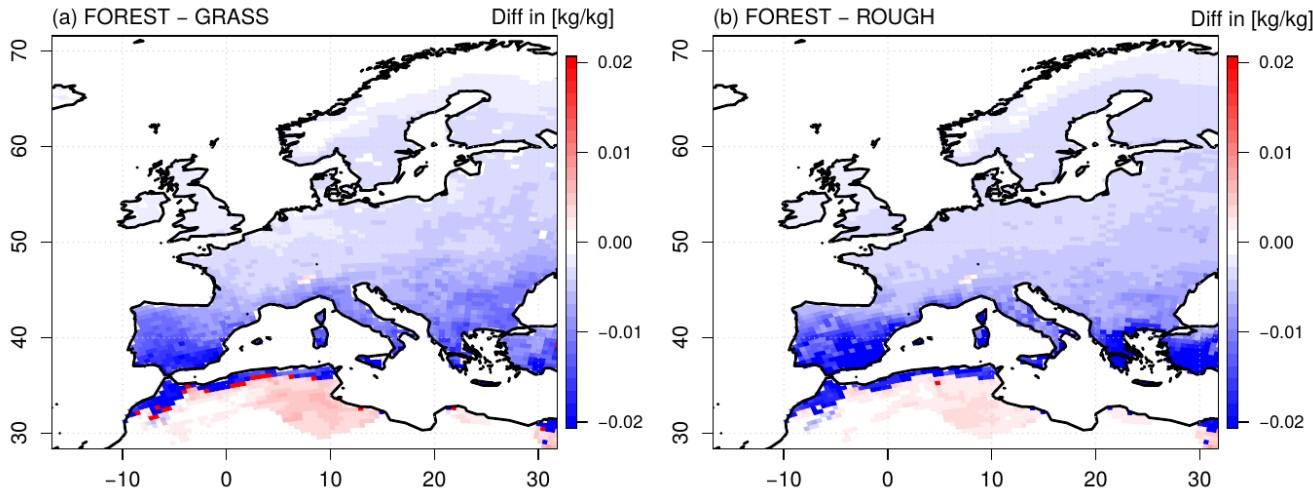

**Figure 4:** Differences in mean saturation deficit [in kg water vapor per kg wet air] between the vegetation and the atmosphere in summer between the FOREST and the GRASS experiment (a), and the FOREST and the ROUGH experiment (b), over the simulation period 1986-2015. The saturation deficit is calculated for the daily maximum surface temperature (top of vegetation).

In Southern Europe, the reduction of the saturation deficit is particularly pronounced. As a result, the reduced saturation deficit exceeds the impact of the increased transfer coefficient in the transpiration flux calculation (Eq. 1) and evapotranspiration is reduced. In Northern Europe, on the contrary, the reduction of the saturation deficit in the FOREST simulation is less pronounced. As shown in Figure 1, Northern Europe is completely covered by coniferous forest in the FOREST simulation. Coniferous forest has a high *LAI* and low albedo values and thus, low $r_c$ and high $c$ values. In Northern Europe, a slightly reduced saturation deficit is consequently facing a high transfer coefficient. This higher transfer coefficient therefore exceeds the impact of the reduced saturation deficit in the flux calculation (Eq. 1) and evapotranspiration is increased. In Central Europe, the saturation deficit in the FOREST run is comparable to Northern Europe. But in contrast to Northern Europe, regions of increased evapotranspiration are simulated as well as regions of reduced evapotranspiration compared to the GRASS simulation (Figure 2a). As already mentioned in section 3.1, the regions of increased evapotranspiration coincide with regions covered by coniferous forests, while regions of reduced evapotranspiration are covered by deciduous forests. Since the saturation deficit reduction in the FOREST run is comparable for both forest types in Central Europe (Figure 4a), these different evapotranspiration responses to afforestation must be associated with differences in the transfer coefficient $c$ (Eq. 1). The transfer coefficient $c$ of coniferous forest must therefore be higher than the one of deciduous forest. In a coniferous forest *LAI* is increased and albedo is reduced in comparison to a deciduous forest, while in deciduous forest the root depth and $c_{veg}$ are increased. Thus, both forest types have characteristics which lead to high $c$ values. However, since evapotranspiration in Central Europe is higher for coniferous forests than for deciduous forests, the impact of *LAI* and the albedo (pronounced in coniferous forests) on $c$ must be higher than the impact of the root depth and $c_{veg}$ (pronounced in deciduous forests). As a result, the impact of the higher transfer coefficient $c$ of coniferous forests surpasses the effects of the

lower saturation deficit in Central Europe in the transpiration flux calculation and evapotranspiration is increased, while for deciduous forests the impact of the reduced saturation deficit is dominating and evapotranspiration is reduced.

As described in section 3.1., surface roughness has only a minor impact on the extent of the transfer coefficient $c$. But its effects on the humidity gradients are large. As shown in Figure 4b, the reduction of the surface roughness in the ROUGH simulation, results all over Europe in increased saturation deficits, which are similar to the GRASS run. Thus, the surface roughness is the main driver for the different saturation deficits in FOREST and GRASS. The reasons for this surface roughness effect on the saturation deficits are described in detail in the next section.

### 3.3 Effects of surface roughness

Differences in evapotranspiration as seen for the FOREST and GRASS runs (Figure 2), inevitably affect the atmospheric conditions in these simulations. For instance, the increased evapotranspiration rates in Northern Europe in FOREST lead to an increased cloud cover in this region (Figure 5a). The incoming solar radiation is consequently reduced in comparison to GRASS. However, since the albedo of the trees in the FOREST simulation is lower than the albedo of grassland in the GRASS run, the reduction of the incoming solar radiation is compensated and net short-wave radiation is slightly increased in Northern Europe (Figure 5b). For the rest of the European continent, this albedo effect is even stronger pronounced and the net short-wave radiation is considerably increased, since cloud cover is not changed compared to GRASS. But this increased radiative energy input does not result in higher surface temperatures (Figure 6a; since evapotranspiration mainly takes place during the day, here and in the following, the daily maximum temperatures are considered). All over Europe lower daily maximum surface temperatures are simulated in FOREST than in GRASS. These lower surface temperatures cannot be caused by an evaporative cooling, associated with increased latent heat fluxes as generally supposed (e.g. Bonan, 2008), since at least in Southern and Central Europe evapotranspiration is reduced in FOREST (Figure 2a). As stated by Breil et al., (2020), the lower surface temperatures in FOREST are mainly caused by increased sensible heat fluxes all over Europe (Figure 6b), which transform and transport the increased energy input from the net short-wave radiation into the atmosphere, without increasing the surface temperature.

These increased sensible heat fluxes are induced by the higher surface roughness of a forest compared to grassland, as demonstrated by the results of the ROUGH simulation (Figure 5c-d and Figure 6c-d). Due to the increased evapotranspiration rates in ROUGH in Northern Europe (Figure 2b), cloud cover is increased in this region in comparison to the FOREST run (Figure 5c). The net short-wave radiation is consequently slightly reduced (Figure 5d). But for the rest of the European continent, net short-wave radiation in FOREST and ROUGH is on the same high level, due to the unchanged albedo values. The reduced surface roughness in ROUGH reduces all over Europe the sensible heat transport into the atmosphere (Figure 6d). Thus, the high radiative energy is not as efficiently transformed and transported into the atmosphere as in FOREST, with the consequence that the surface temperatures are increased, similarly to the GRASS simulation (Figure 6c).

As described in Eq. (1), the saturation deficit between the surface and the atmosphere, depends on the surface temperature. Due to the increased surface roughness of a forest, this surface temperature is reduced compared to grassland. As a result, the saturation deficit of forest to the atmosphere is lower than for grassland (Figure 4a). Finally, this leads in Southern and Central Europe to a lower forest evapotranspiration (Figure 2a). Thus, the lower surface temperatures of forests compared to grassland are there not a result of evaporative cooling, but of the increased surface roughness. These lower surface temperatures, in turn, then even decrease forest evapotranspiration.

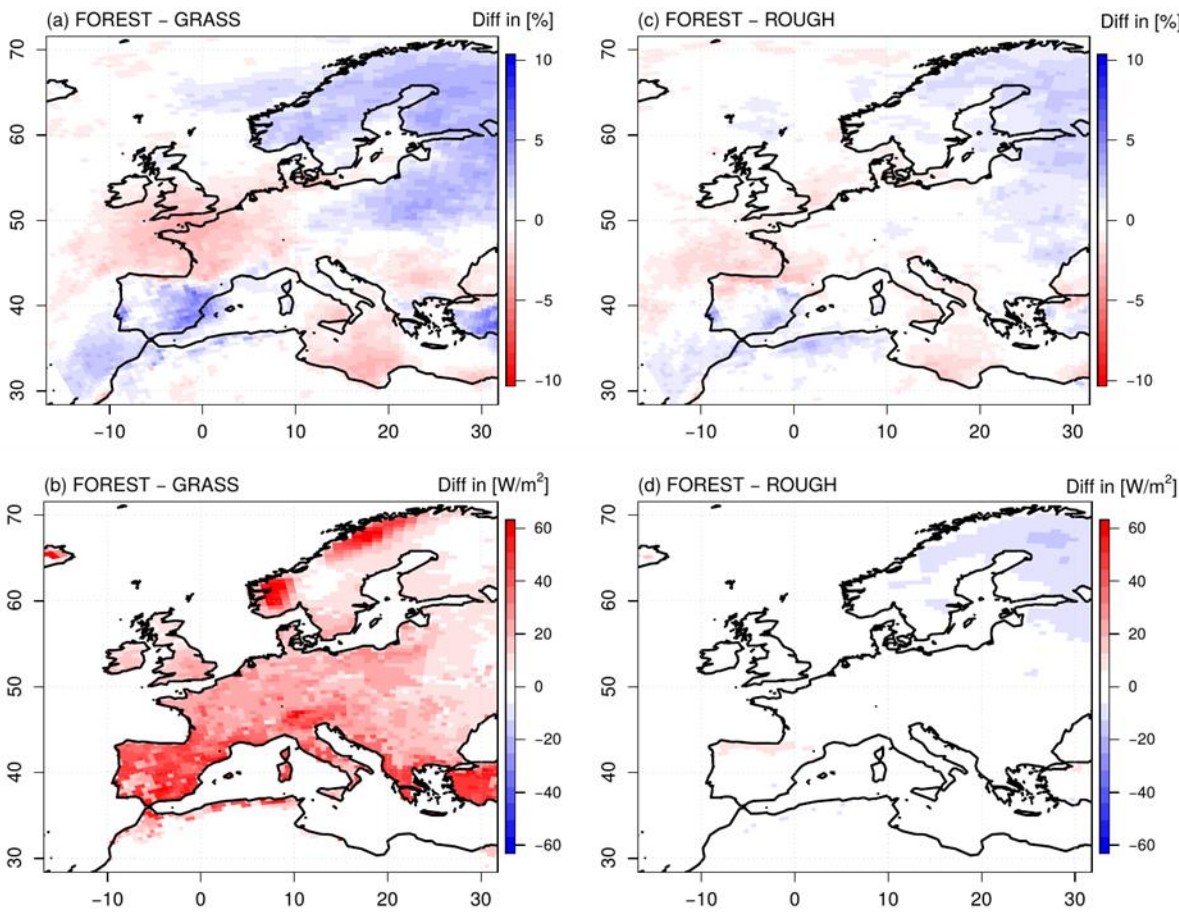

**Figure 5:** Differences in mean seasonal cloud cover (a,c), net short-wave radiation (b,d), in summer between the FOREST and the GRASS experiment (a-b), and the FOREST and the ROUGH experiment (c-d), over the simulation period 1986-2015.

## 4 Discussion and Conclusions

In the framework of idealized regional climate simulations with CCLM-VEG3D for two extreme land use change scenarios (FOREST and GRASS), diverging evapotranspiration responses are simulated. In Northern Europe evapotranspiration is increased with afforestation, in Southern and Central Europe evapotranspiration is decreased. Especially the reduced forest evapotranspiration rates in Southern and Central Europe are in contradiction to the prevailing scientific doctrine that forest evapotranspiration is enhanced (e.g. Bonan, 2008), due to deeper roots (Schenk and Jackson, 2003) and a higher Leaf Area Index (Henderson-Sellers, 1993) than grassland. However, these results qualitatively reflect the varying evapotranspiration rates of forests and grasslands in European summer, documented in numerous observation and modelling studies (Zhang et al., 2001; Williams et al., 2012; Davin et al., 2020).

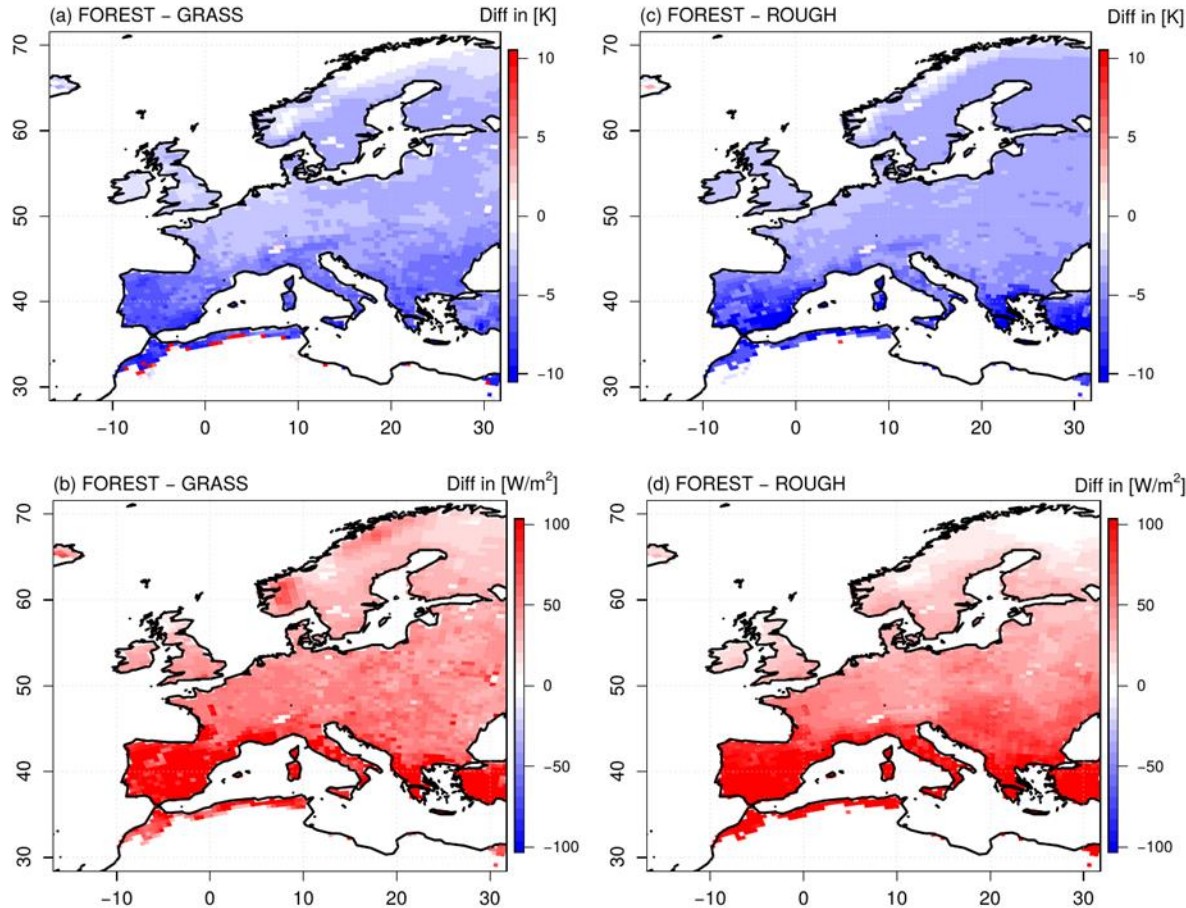

**Figure 6:** Differences in mean seasonal mean daily maximum surface (top of vegetation) temperature (a, c), mean seasonal sensible heat fluxes (b, d) in summer between the FOREST and the GRASS experiment (a-b), and the FOREST and the ROUGH experiment (c-d), over the simulation period 1986-2015.


Climate simulations with incorporated Land Surface Models (LSMs) are an appropriate method to analyze the reasons for these varying evapotranspiration rates of forests and grasslands. However, models constitute only a simplified description of reality and thus, cannot represent the complex biogeophysical processes in nature comprehensively. For instance, VEG3D does not consider the effects of the multilayer canopy structure of trees (effects of shaded and unshaded leaves; Bonan et al.,

2012) or the influence of the understory on evapotranspiration rates, which can contribute substantially to total evapotranspiration in forests (e.g. Yepez et al., 2003). Furthermore, VEG3D does not consider the impact of temperature and vapor pressure deficit on stomata closure. But the results of model-intercomparison studies show that more sophisticated LSMs, in which these biogeophysical effects are integrated, exhibit comparable evapotranspiration responses to afforestation as VEG3D (e.g. de Noblet-Ducoudré et al. 2012; Davin et al., 2020). For instance, in the framework of the LUCAS project,

simulations with the classic model VEG3D and the more sophisticated Community Land Model under the same atmospheric boundary conditions, show similar spatial patterns of increased or reduced evapotranspiration rates with afforestation (Davin et al., 2020). Thus, the differences in the model complexity (effects of shaded and unshaded leaves or the vapor pressure dependency of stomata closure) cannot be the main reason for the simulated differences in evapotranspiration responses of forests and grasslands. These different evapotranspiration responses must rather be caused by a fundamental mechanism, which is simulated in both, classic as well as complex LSMs. In order to get to the bottom of these fundamental processes,


the use of a less complex model can even be beneficial. In such a model, the degrees of freedom are reduced and functional interrelations can consequently be deduced more easily. Therefore, by means of a sensitivity study with this less complex CCLM-VEG3D model, in which the surface roughness of forests was reduced to grassland (ROUGH), this fundamental mechanisms behind the varying evapotranspiration rates of forests and grasslands could be clearly revealed:

Due to a higher surface roughness, the daily maximum surface temperatures (top of the vegetation) of a forest are lower than of grassland (Breil et al., 2020). The saturation deficit between the vegetation and the atmosphere (1), which depends on these surface temperatures (Eq. 1), is consequently reduced and counteracts the transpiration facilitating characteristics of a forest ((2), high transfer coefficient due to deep roots, high LAI, low albedo). Therefore, the question whether forests or grasslands transpire more water, depends on the balance between the two factors (1) and (2).

The simulation results show that the interplay of these two forces depends, on the one hand, on the latitude. In the Southern Europe, with its intense solar radiation, the surface temperature is strongly increasing, if energy is not efficiently transformed into sensible heat fluxes by turbulent processes. Due to its low surface roughness, grassland is not able to transform the solar energy as efficient as forest. The surface temperature and thus also the saturation deficit (1) is consequently stronger increased than for forest. The impact of factor (1) therefore exceeds the effects of factor (2) and grassland transpiration is increased

compared to forest. In Northern Europe, on the contrary, the incoming solar radiation is lower. Thus, the surface temperature differences and saturation deficits between forest and grassland (1) are not as pronounced as in the southern parts of Europe. The impact of factor (2) surpasses consequently the effects of factor (1) and forest transpiration is increased compared to grassland. The dependency of the evapotranspiration rates of forests and grasslands on the latitude is also documented in satellite observations (e.g. Li et al., 2015). In this context, the simulated increase in evapotranspiration with afforestation for

large parts of Central and Northern Europa are in line with observations (e.g. Duveiller et al., 2018), while the simulated reduction in evapotranspiration in the Mediterranean is not reflected by observations (e.g. Rohatyn et al., 2018). One potential explanation for these deviations between the CCLM-VEG3D model results and observations is the missing consideration of summertime senescence of grasslands in Mediterranean regions and the associated reduction in grassland evapotranspiration (Ryu et al., 2008). Another possible reason for the disagreement between the simulation results and the observations is the

missing consideration of vapor pressure effects on the stomatal resistance in CCLM-VEG3D. For instance, in Southern Europe the saturation deficit of forests is particularly lower than for grasslands. In contrast to the simulated trees in CCLM-VEG3D, real trees are potentially able to adapt to this lower saturation deficit, by further reducing the stomatal closure and thus the transfer coefficient. In line with the introduced evapotranspiration concept, the transpiration facilitating characteristics of forests (2) would be further enhanced, counteracting the reduced saturation deficit (1) in Southern Europe and thus, would

increase forest evapotranspiration.

   On the other hand, the simulation results show that the balance between factor (1) and (2) is differently pronounced for different forest types. In Central Europe, for instance, deciduous and coniferous forests are showing opposing evapotranspiration responses to afforestation, although they are facing a comparable saturation deficit (1). Differences in the evapotranspiration rates must consequently be associated with differences in the transfer coefficients (2). A deciduous forest,

for instance, has a lower LAI and higher albedo values than a coniferous forest (Table 2). The transfer coefficient is consequently lower and factor (2) is becoming weaker. The impact of the saturation deficit (1) is therefore dominating the effects of factor (2) and the transpiration rates of deciduous forests are reduced compared to grassland in Central Europe. But for coniferous forest, which are facing a similar saturation deficit (1), the impact of factor is increased (2), due to their higher LAI and lower albedo values. The transpiration rates are consequently higher for coniferous forests in this region. These

results are also in line with observation-based studies, showing that evapotranspiration rates differ between different forest types (e.g. Brown et al., 2005), whereby higher evapotranspiration rates are generally assigned to coniferous forests (e.g. Teuling, 2018). Furthermore, Marc and Robinson, (2007) showed that also the age of the forest affects evapotranspiration.

In this study, only the results of model simulations are presented, which obviously depend on the used parameterizations and parameters. In the specific CCLM-VEG3D setup, for instance, only two different forest types (coniferous and deciduous) are

applied, which might not completely represent the whole variety of European forests. Generalizations, as well as under- or overestimations of certain physical processes can locally result. Therefore, this study does not claim for general validity. The transpiration rates of forests and grasslands depend on the weighting of the respective factors (1) and (2). Since this weighting is model-specific, slightly different evapotranspiration responses of forests and grasslands are anticipated for different model simulations. Moreover, different evapotranspiration responses can also be expected within observational data, since the

biogeophysical characteristics of forests and grasslands vary also in nature (Garratt, 1993; Henderson-Sellers, 1993; Schenk and Jackson, 2003). Taking these uncertainties into account differences between the CCLM-VEG3D results and observations, as present in Southern Europe (Rohatyn et al., 2018), can potentially be explained.

However, it is generally difficult to assess the effects of afforestation by a direct comparison of the CCLM-VEG3D model results with observational data, due to discrepancies on the scale of processes considered in models and observations (Davin

et al., 2020). In observational data (satellite data as well as data from eddy covariance flux towers) forests and grasslands in immediate vicinity are compared. Differences in the measured fluxes are therefore directly related to the local land cover differences (Bright et al., 2017). In contrast, differences in model results for forests and grasslands are additionally affected by large-scale atmospheric feedback processes (Winckler et al., 2017). Therefore, it is difficult to assess the CCLM-VEG3D model results quantitatively and qualitatively in comparison to observations. Thus, with this study, it is not intended to answer

the question whether in specific regions observation-based studies are correct which show higher evapotranspiration rates of forests (e.g. Zhang et al., 2001; Li et al., 2015; Chen et al., 2018; Duveiller et al., 2018), or studies which document the opposite behavior (e.g. Wicke and Bernhofer, 1996; Teuling et al., 2010; Williams et al., 2012). In this study, rather a mechanism is presented that explains how these different transpiration responses of forests and grasslands can generally evolve in Europe and by which factors they are controlled. In this context, especially an explanation for the hardly

comprehensibly lower evapotranspiration rates of forests during summer, can be provided in a physically consistent way.

*Code availability.* The coupled version of COSMO-CLM and VEG3D is available upon request from Marcus Breil.

*Author contributions.* MB designed the study. MB performed the model simulations and analysis. All authors (MB, ELD, DR) contributed to writing and revising the manuscript.

*Competing interests.* The authors declare that they have no conflict of interest.

*Acknowledgements.* LUCAS is supported by WCRP CORDEX as a Flagship Pilot Study. MB acknowledges funding by the Federal Ministry of Education and Research in Germany (BMBF) under the ClimXtreme research program. ELD acknowledges support from the Swiss National Science Foundation (SNSF) through the CLIMPULSE project. All simulations were performed at the German Climate Computing Center (DKRZ).

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
