# Peer review of "What determines the sign of the evapotranspiration response to afforestation in the European summer?"

_Biogeosciences, 2020_

## Referee Comment (RC1) · Anonymous Referee #1 · 10 Sep 2020

What determines the sign of the evapotranspiration response to afforestation in the European summer? Marcus Breil, Edouard L. Davin and Diana Rechid

BioGeoScience (2020 – 275) paper:

The paper deals with the afforestation effect on evapotranspiration rate (ET) of the European continent. The paper uses a Regional Climate Model COSMO-CLM to compare ET changes due to a scenarios of afforestation of the whole European landscape vs. the ET rate of a fully grassy European landscape. Five different variables, that are dependent on three land cove types (two forests types and grassland) are used in the model to deduce the ET rate per unit area for the continent. The model finds that

what mainly governs the ET rate in the summer time is the water saturation difference between the ecosystem surface and the above air. In southern Europe, where solar radiation burden is high, grassland ecosystem ET is higher than forest ET because the grassland surface temperature is higher than that of the forest ecosystems, thus the water deficit there is higher. In northern Europe, forests ET is higher and this due to higher absorb radiation by the forest ecosystem, while a small surface temperature difference exists between the different ecosystem types. It is an interesting, conceptual paper that tries to help resolving an ongoing question of the effect of land cover change on ecosystems ET rate, in particularly by the change from a grassland to a forest ecosystem across a wide climatic conditions. As such the paper is within the scope of the journal and of high interest for wide disciplinary communities. However, I find two major weak points in the paper that require serious revisions: 1. Model results vs. ground base measurements results. As the authors rightly wrote, based mainly on runoff measurements, forest ecosystems ET are mostly higher than grass ecosystems ET and the differences are functions of many variables, partially presented by the authors. Based on what I am familiar with, in most (if not all) Mediterranean dryer parts, summer ET in forest is higher than that of any paired grasslands sites. See, for example, papers on California (Ryu, et al., 2008, and Baldocchi et al., 2009) and for the Eastern Mediterranean region (Rohatyn, 2018), which seem not to agree with the paper main results. An important part of the explanation for the lower ET in grassland ecosystems in summer in such regions, is that the grassland is mainly annuals, which are dying toward the summer while the trees keep evaporating all year long. This is likely the adaptation of annuals grassland plant types to the regional dry climatic conditions. In wetter regions, the ET difference, based on FluxNet data, are less pronounced, and the paper is in agreement with studies that show that the ET differences depends on local conditions. This leads to the next comments.

2. Comment for the conceptual aspects. a. As the Authors rightly mention, vegetative ecosystem is much more complicated than described by the 5 parameters present in Table 1. However, it seems, there are several important mechanisms that could

override the dominant effect of the increase in water saturation deficit presented by the paper. Ranking the importance of the different mechanisms, function of the local climatic conditions, on plants types, its ages, its density, soil conditions, are avoided. Among those important factors, there is insufficient consideration in the paper of factors such as: the phenology effects (e.g., the annuals life span; see above), the structural effects on the transpiration rate (trees are multi–layers, which has an effect on the leaf to air temperature difference and VPD within the canopy, on light intensity, and more), the understory contribution to the ecosystem ET, etc. Obviously, the model cannot include all of these effects, but should at least be discussed, with respect to the difference between the model finding and measurements results, and to provide possible explanations, and possibly how to better simulate these additional factors. b. Feedbacks between the vegetation and the atmosphere. It should be possible for a paper, where the results are based on a regional climatic model (COSMO), to discuss some vegetation-atmosphere feedbacks. For example, it is shown that the sensible heat flux is higher at the southern parts of the continent, this should dry the air and raises its temperature and may increase the leaf to air VPD for the forest model runs. Or, what is the effect of the higher ET (by the grass) on cloudiness and Rn? Referring to such effects could be of a valuable to such model-based paper.

Minor comments: 1. Since the effect of higher ET by forest is a puzzle for most readers and the explanation is through the higher surface temperature of the grass ecosystem, it is suggested to move this text to an earlier part of the results section, including Fig.' 5 b & e. Does the model calculate the leaves' skin temperature, and if so, how? 2. The paragraph, starting in line. 163 is unclear. 3. Line 182. It is likely that soil ET rate is affected by soil layers deeper than 5 cm. This sentence is questionable. And for line 187 - the soil contribution to ET could be very important (up to several ten percent of total ET). 4. Figure 3, units for the soil humidity values are unclear. Also note that part 'c' is noted twice in the caption (instead of 'd'). 5. Figure 4 units are unclear. 6. To better understand the different effecting parameters on ra and rc between the ecosystems types it is suggesting to add Wwilt and Wroot values to table 2.

Papers: Rohatyn, S., et al. (2018). "Differential Impacts of Land Use and Precipitation on "Ecosystem Water Yield"." Water resources research 54(8): 5457-5470.

Baldocchi Dennis, Qi Chen, Xingyuan Chen, Siyan Ma, Gretchen Miller, Youngryel Ryu, Jingfeng Xiao, Rebecca Wenk and John Battles (2009). "The Dynamics of Energy, Water and Carbon Fluxes in a Blue Oak (Quercus douglasii) Savanna in California, USA", in: "Ecosystem Function in Global Savannas: Measurement and Modeling at Landscape to Global Scales" – edited by Michael J. Hill and Niall P. Hanan, CRC/Taylor and Francis.

Ryu Youngryel, Dennis D. Baldocchi, Siyan Ma and Ted Heh (2008), "Interannual variability of evapotranspiration and energy exchange over an annual grassland in California", JOURNAL OF GEOPHYSICAL RESEARCH, VOL. 113.

―――――――――――――――――――――

---

## Referee Comment (RC2) · Anonymous Referee #2 · 28 Sep 2020

The authors have identified an important, and poorly understood, aspect of the effects of afforestation/deforestation in temperate latitudes on climate: do forests increase or decrease evapotranspiration (ET) compared with grasslands? Some observational studies suggest forests have greater rates of ET; some show the opposite. Many modeling studies show forests increase ET; others do not. The topic is fraught with confusion. I had hoped this manuscript would clarify the science and provide a strong, insightful understanding of forest ET and the factors controlling ET. However, by using a poorly documented model, and by not adequately describing the model, the rationale for parameterizations and parameter values, and the limitations of the model, the manuscript does not clarify the science and, instead, adds more confusion to the literature.

1. I have several concerns about the VEG3D model. From the few equations given in the manuscript, it appears to be a highly simplified land surface model. There is nothing wrong with that! But I suspect the findings of the study do not extend to more complex land surface models. The authors need to provide a thorough description of the model, justify the parameterizations used in the model, and justify parameter values. They also need to discuss how the simplifications of VEG3D might limit the generality of the results.

This is not the first time this issue has arisen. VEG3D was used in a previous study by the lead author:

Breil, M., and Coauthors, 2020: The Opposing Effects of Reforestation and Afforestation on the Diurnal Temperature Cycle at the Surface and in the Lowest Atmospheric Model Level in the European Summer. J. Climate, 33, 9159–9179

In full disclosure, I was a reviewer of that manuscript and noted in my review that VEG3D is a poorly documented model, is not widely known by the scientific community, and has not been tested in temperate forest/grassland simulations in comparison with flux tower measurements. That does not mean that the model is deficient or inappropriate for this study, but the description of ET provided in the current manuscript reveals some non-standard formulations in the model that likely limit the generality of the results.

1a. The authors describe the aerodynamic resistance ra used in the transpiration equation (eq 3). This is not a standard formulation of aerodynamic resistance (I have never seen it before). The resistance depends on wind speed at the top of the canopy, leaf area index, and some undescribed parameters. Classic textbooks on micrometeorology and boundary layer meteorology formulate the resistance using integrated flux-profile relationships between the apparent source/sink in the canopy (at a height equal to the roughness length [z0] plus displacement height [d]) and the lowest model level

in the atmosphere [z]:

ra = [ln(z-d)/z0]^2 / (k^2 * u)

u is wind speed at z. Depending on the specific model, z0 can be either that for momentum or for scalars, and ra is adjusted for atmospheric stability. What is the justification for eq 3, which seems to go back to two very old papers (Deardorff, 1978; Taconet et al., 1986)? Why is this equation used rather than classic boundary layer theory? It seems from eq 3 that roughness length only enters the model through wind speed at the top of the canopy (uaf), but there is no equation for uaf. It appears to go back to Goudriann's old work. This is very important, because the key outcome of the study is that surface roughness is the primary difference between forests and grasslands. Readers must understand precisely how surface roughness is used in the model and why particular formulations are used in the model.

1b. The formulation of canopy resistance to transpiration (eq 4) is also somewhat odd. It goes back to an equation in Deardoff (1978), in which canopy resistance depends on a specified minimum resistance that is modified for solar radiation and soil moisture. Most current-generation land surface models use an approach that couples photosynthesis and stomatal conductance through the Farquhar et al. (1980) photosynthesis model and semi-empirical stomatal conductance models such as proposed by Ball-Berry or Medlyn. In addition to light and soil moisture effects on stomatal conductance, those models also include temperature and vapor pressure deficit (VPD) effects on stomatal conductance. The VEG3D model ignore those latter two effects. That exclusion greatly limits the generality of the main finding of the study (that VPD, as modified by surface roughness, is a key determinant of differences in ET between forests and grasslands). The response of stomata to VPD is not considered (i.e. stomata close as VPD increases). Nor are the indirect effects of VPD on stomata through leaf temperature considered. Again, readers need to know why eq 4 is used in contrast with more common stomatal conductance models and what the implications of eq 4 are for the main findings of the study.

1c. The term (1+0.5*LAI)/LAI is common to both ra and rc. What does this term represent? It seems to be a scaling term for canopy LAI (i.e, from a leaf resistance to a canopy resistance). Aerodynamic resistance is commonly expressed per unit ground area. Why does ra need to be scaled by LAI?

2. The authors emphasize that differences between forest and grassland arise in terms of five model parameters: surface roughness, albedo, root depth, leaf area index, and minimum stomatal conductance.

2a. The justification for several parameter choices goes back to papers by Garratt (1993) and Henderson-Sellers (1993). There has been a lot of model development since then. How do these parameter choices compare with values used in the current generation of land surface models?

2b. Table 2 shows only a small difference in rmin between forest and grassland, and no difference between coniferous and deciduous forest. What is the justification for the parameter values? Are there physiological measurements that support them? The values for rmin are very important to the results of the study. The relative contributions of aerodynamic resistance and canopy resistance to total resistance determine the model sensitivity to roughness length. The fact that rmin is similar for all vegetation precludes physiological differences in stomatal conductance from determining differences in ET.

2c. No details are given on how root depth affects transpiration, or how the root depth parameter is used in the model. The root depth of deciduous forest is twice that of coniferous forest. Is this the reason for the differences between deciduous and coniferous forests when they are converted to grassland?

2d. No details are given for albedo. What is the radiative transfer parameterization in the model? Land surface models typically simulate radiative transfer for visible and near-infrared wavebands and for direct and diffuse radiation. Albedo is a complex result of leaf and stem reflectances, leaf and stem area index, solar zenith angle, and soil moisture. Because only a single albedo is listed as a parameter in Table 2, this

makes me think there is no such complex radiative transfer parameterization in VEG3D and instead the model uses a bulk surface albedo that is prescribed as a parameter. Readers need further information.

3. A striking aspect of Figure 2 is the difference between coniferous and deciduous forests when replaced with grassland. Summer latent heat fluxes are larger in coniferous forest compared with grassland but are smaller than grassland in deciduous forest. This pattern is universally consistent throughout the domain, except for southern Spain and Turkey (smaller latent heat fluxes compared with grassland in a mostly coniferous forest region). The authors acknowledge the influence of forest type (lines 165-166), but for the most part discuss their results in terms of Northern Europe versus Southern/Central Europe. For example, the authors frame their conclusions as: "In Northern Europe evapotranspiration is increased with afforestation, in Southern and Central Europe evapotranspiration is decreased" (lines 261-262). The differing results of coniferous and deciduous forests are not even mentioned in the abstract. I would like to see more of a discussion of coniferous versus deciduous forests.

3a. What, specifically, are the differences between these forests that cause the results? One generally thinks of coniferous forests as having a more conservative water-use strategy than deciduous forests (seen, for example, in higher stomatal resistance). But both forests have the same minimum resistance. Is the different response related to rooting depth? In their analysis of soil water (Figure 3), the authors suggest it is not but the analysis is not definitive. It would be better to look at the soil moisture stress term in canopy resistance.

3b. Are the results consistent with observations? What do flux towers show? What does MODIS ET show (but remember that MODIS ET is a modeled product).

4. The crux of the study is Figure 4, which shows the difference in saturation deficit between the forest and grassland simulations. Saturation deficit decreases for forests throughout Europe, with a particularly large decrease in latitudes south of about 40N.

The authors discuss the results in light of VPD, resistances, and other parameters that affect transpiration (lines 219-234). No data or figures are provided to justify the interpretation. Skeptical readers need to see more evidence that supports the argument if they are to believe the study.

5. Figure 5d: Why does net shortwave radiation change when roughness length is changed to that of grassland?

6. Lines 288-290: The authors state that "the dependency of the evapotranspiration rates of forests and grasslands on the latitude is also documented in satellite observations (e.g. Li et al., 2015), showing for example higher evapotranspiration rates of grasslands in South-Eastern Europe, while in Central and Northern Europe evapotranspiration is lower than in forests (Duveiller et al., 2018)". Li et al used MODIS ET, which is a modeled product. What did Duveiller et al base their analysis on? And, remember, that the more striking aspect of Figure 2 is not the latitudinal dependence but the difference between coniferous and deciduous forests. What do observations say about that difference?

---

## Author Comment (AC1) · 23 Oct 2020

The paper deals with the afforestation effect on evapotranspiration rate (ET) of the European continent. The paper uses a Regional Climate Model COSMO-CLM to compare ET changes due to a scenarios of afforestation of the whole European landscape. Five different variables, that are dependent on three land cove types (two forests types and grassland) are used in the model to deduce the ET rate per unit area for the continent. The model finds that what mainly governs the ET rate in the summer time is the water saturation difference between the ecosystem surface and the above air. In southern Europe, where solar radiation burden is high, grassland ecosystem ET is higher than forest ET because the grassland surface temperature is higher than that of the forest ecosystems, thus the water deficit there is higher. In northern Europe, forests ET is higher and this due to higher absorb radiation by the forest ecosystem, while a small surface temperature difference exists between the different ecosystem types. It is an interesting, conceptual paper that tries to help resolving an ongoing question of the effect of land cover change on ecosystems ET rate, in particularly by the change from a grassland to a forest ecosystem across a wide climatic conditions. As such the paper is within the scope of the journal and of high interest for wide disciplinary communities. However, I find two major weak points in the paper that require serious revisions:
- Thanks for your assessment. We hope that we are able to respond satisfactorily to your comments and clear the open issues you raised.

1. Model results vs. ground base measurements results. As the authors rightly wrote, based mainly on runoff measurements, forest ecosystems ET are mostly higher than grass ecosystems ET and the differences are functions of many variables, partially presented by the authors. Based on what I am familiar with, in most (if not all) Mediterranean dryer parts, summer ET in forest is higher than that of any paired grasslands sites. See, for example, papers on California (Ryu, et al., 2008, and Baldocchi et al., 2009) and for the Eastern Mediterranean region (Rohatyn, 2018), which seem not to agree with the paper main results. An important part of the explanation for the lower ET in grassland ecosystems in summer in such regions, is that the grassland is mainly annuals, which are dying toward the summer while the trees keep evaporating all year long. This is likely the adaptation of annuals grassland plant types to the regional dry climatic conditions. In wetter regions, the ET difference, based on FluxNet data, are less pronounced, and the paper is in agreement with studies that show that the ET differences depends on local conditions. This leads to the next comments.
- We agree, that a lot of observation-based studies indicate higher ET rates of forests in comparison to grasslands in Mediterranean regions. In the revised manuscript, this issue is pointed out more clearly.

Lines (346-351):
"In this context, the simulated increase in evapotranspiration with afforestation for large parts of Central and Northern Europa are in line with observations (e.g. Duveiller et al., 2018), while the simulated reduction in evapotranspiration in the Mediterranean is not reflected by observations (e.g. Rohatyn et al., 2018). One potential explanation for these deviations between the CCLM-VEG3D model results and observations is the missing consideration of summertime senescence of grasslands in Mediterranean regions and the associated reduction in grassland evapotranspiration (Ryu et al., 2008)."

However, a direct comparison of observational data with our model results is difficult, due to the different spatial scales of the data. While observational data reflect the local differences between forest and grassland transpiration rates, in our simulation setup, large-scale forestation scenarios are

applied to analyze the general transpiration responses to forestation in an idealized and isolated way. It is therefore very difficult to assess the model results quantitatively and qualitatively.

Lines (379-384)
"However, a direct comparison of the CCLM-VEG3D model results with observational data is generally difficult, due to the different spatial representativity of the data. While observational data (satellite data as well as data from eddy covariance flux towers) reflect the local transpiration responses to forestation (Bright et al., 2017), in the CCLM-VEG3D simulation setup, large-scale forestation scenarios are applied to analyze the general transpiration responses to forestation in an idealized and isolated way. Therefore, it is difficult to assess the CCLM-VEG3D model results quantitatively and qualitatively in comparison to observations"

Furthermore, the aim of the study is not to reproduce observed transpiration rates. We rather want to understand the reason for the contradicting evapotranspiration responses to forestation existing in observations and model results. In this context, we are able to introduce a physically consistent explanation for this phenomena, in which the evapotranspiration responses are described as an interplay of two factors, namely the reduced vapor pressure deficit in forests facing their evapotranspiration facilitating biogeophysical characteristics. Since the weighting of both factors is differently pronounced in each model, and furthermore, depends on latitude and forest type, deviating evapotranspiration responses are observed and simulated. Thus, in comparison to observations, it seems that in our model the weighting of both factors is not absolutely correct for the Mediterranean (as far as we can assess it, regarding the different spatial scales). This aspect is also further emphasized in the revised manuscript

Lines (374-378):
"Since this weighting is model-specific, slightly different evapotranspiration responses of forests and grasslands are anticipated for different model simulations. This can also be expected for observed evapotranspiration rates, since the biogeophysical characteristics of forests and grasslands vary also in nature (Garratt, 1993; Henderson-Sellers, 1993; Schenk and Jackson, 2003), potentially explaining differences between the CCLM-VEG3D results and observations, especially in Southern Europe (Rohatyn et al., 2018)."

2. Comment for the conceptual aspects.
a. As the Authors rightly mention, vegetative ecosystem is much more complicated than described by the 5 parameters present in Table 1. However, it seems, there are several important mechanisms that could override the dominant effect of the increase in water saturation deficit presented by the paper. Ranking the importance of the different mechanisms, function of the local climatic conditions, on plants types, its ages, its density, soil conditions, are avoided. Among those important factors, there is insufficient consideration in the paper of factors such as: the phenology effects (e.g., the annuals life span; see above), the structural effects on the transpiration rate (trees are multi–layers, which has an effect on the leaf to air temperature difference and VPD within the canopy, on light intensity, and more), the understory contribution to the ecosystem ET, etc. Obviously, the model cannot include all of these effects, but should at least be discussed, with respect to the difference between the model finding and measurements results, and to provide possible explanations, and possibly how to better simulate these additional factors.
- you are right, VEG3D does not include these effects and thus, is not able to reflect the whole complexity of the soil-vegetation-atmosphere system. In the revised manuscript these model deficiencies and their potential impact on the differences to observations are discussed in more detail.

Lines (314-319):
"Climate simulations with incorporated Land Surface Models (LSMs) are an appropriate method to analyze the reasons for these varying evapotranspiration rates of forests and grasslands. However, models constitute only a simplified description of reality and thus, cannot represent the complex biogeophysical processes in nature comprehensively. For instance, VEG3D does not consider the effects of the multilayer canopy structure of trees (effects of shaded and unshaded leaves; Bonan et al., 2012) or the influence of the understory on evapotranspiration rates, which can contribute substantially to total evapotranspiration in forests (e.g. Yepez et al., 2003)."

Lines (348-351):
"One potential explanation for these deviations between the CCLM-VEG3D model results and observations is the missing consideration of summertime senescence of grasslands in Mediterranean regions and the associated reduction in grassland evapotranspiration (Ryu et al., 2008)."

b. Feedbacks between the vegetation and the atmosphere. It should be possible for a paper, where the results are based on a regional climatic model (COSMO), to discuss some vegetation-atmosphere feedbacks. For example, it is shown that the sensible heat flux is higher at the southern parts of the continent, this should dry the air and raises its temperature and may increase the leaf to air VPD for the forest model runs. Or, what is the effect of the higher ET (by the grass) on cloudiness and Rn? Referring to such effects could be of a valuable to such model-based paper.
- you are right. Since sensible heat fluxes are increased in the FOREST simulation, air temperatures are increased and in this way also the capability of the atmosphere to carry water vapor (Breil et al., 2020). But due to the intense vertical mixing within the boundary layer and the associated increased heat capacity of the atmosphere in comparison to the surface, the warming of the atmosphere is less pronounced than the cooling of the surface in the FOREST simulation. The vapor pressure deficit is consequently all over Europe reduced in FOREST, although sensible heat fluxes are increased. Furthermore, we agree that evapotranspiration changes can affect the cloud cover and thus, the net short-wave radiation. This feedback is now discussed in detail in the revised manuscript.

Lines (266-273):
"Differences in evapotranspiration as seen for the FOREST and GRASS runs (Figure 2), inevitably affect the atmospheric conditions in these simulations. For instance, the increased evapotranspiration rates in Northern Europe in FOREST lead to an increased cloud cover in this region (Figure 5a). The incoming solar radiation is consequently reduced in comparison to GRASS. However, since the albedo of the trees in the FOREST simulation is lower than the albedo of grassland in the GRASS run, the reduction of the incoming solar radiation is compensated and net short-wave radiation is slightly increased in Northern Europe (Figure 5b). For the rest of the European continent, this albedo effect is even stronger pronounced and the net short-wave radiation is considerably increased, since cloud cover is not changed compared to GRASS. But this increased radiative energy input does not result in higher surface temperatures"

Lines (282-285):
"Due to the increased evapotranspiration rates in ROUGH in Northern Europe (Figure 2b), cloud cover is increased in this region in comparison to the FOREST run (Figure 5c). The net short-wave radiation is consequently slightly reduced (Figure 5d). But for the rest of the European continent, net short-wave radiation in FOREST and ROUGH is on the same high level, due to the unchanged albedo values."

**Minor comments:**
1. Since the effect of higher ET by forest is a puzzle for most readers and the explanation is through the higher surface temperature of the grass ecosystem, it is suggested to move this text to an earlier

part of the results section, including Fig.5 b & e. Does the model calculate the leaves' skin temperature, and if so, how?

- Thanks for your suggestion, but we would like to maintain the current structure to keep the logical order of the manuscript.

Yes. The leaf temperature is calculated by solving the energy balance of the vegetation layer iteratively.

2. The paragraph, starting in line. 163 is unclear.

- paragraph is rephrased.

"In Southern and Central Europe, evapotranspiration is reduced in the FOREST run compared to the GRASS simulation (Figure 2a). The evapotranspiration reduction in FOREST is in this context particularly strong in Southern Europe. But in Northern Europe the opposite is the case and evapotranspiration is increased in FOREST. In Central Europe, regions with reduced evapotranspiration rates in FOREST coincide with regions covered by deciduous forest (Figure 1). This indicates that differences in evapotranspiration rates between forests and grassland are affected by the prevailing forest type in a region. Thus, the different vegetation characteristics (a-f) of deciduous and coniferous forest, must have an impact on the intensity of the evapotranspiration response to afforestation. But since both forest types have lower resistance values (higher $c$ values) than grasslands, both forest types should also stronger promote transpiration than grasslands, which seems to be in contradiction to the reduced evapotranspiration rates of deciduous forests in Central Europe. Therefore, the resistance values of the different forest types cannot solely explain the opposing transpiration signals."

3. Line 182. It is likely that soil ET rate is affected by soil layers deeper than 5 cm. This sentence is questionable. And for line 187 - the soil contribution to ET could be very important (up to several ten percent of total ET).

- we agree, it is possible that soil depths deeper than 5 cm can be affected by soil evaporation, but the contribution is decreasing with depth. A depth of 5 cm is therefore a meaningful reference to evaluate the contribution of soil evaporation to total evapotranspiration. Furthermore, you are right, in general, the contribution of soil evaporation to total evapotranspiration can be very important. Both statements are therefore specified in the revised paper.

Lines (199-201):
"Differences in the upper 5 cm of the soil (Figure 3b) are used as an indicator for differences in the soil evaporation, since this process is executed through the soil surface (although soil evaporation can also be affected by soil depths deeper than 5 cm)."

Lines (205-206):
"The contribution of soil evaporation to total evaporation is therefore low in both simulations"

The important message of this comparison is that the contribution of soil evaporation to the total evapotranspiration does not differ between FOREST and GRASS and, therefore, differences in the evapotranspiration rates must be caused by differences in the transpiration rates.

4. Figure 3, units for the soil humidity values are unclear. Also note that part 'c' is noted twice in the caption (instead of 'd').

- units are changed in [%] and the caption is revised.

5. Figure 4 units are unclear.

- units are clarified in the caption.

6. To better understand the different effecting parameters on ra and rc between the ecosystems types it is suggesting to add Wwilt and Wroot values to table 2.
- $W_{wilt}$ is the permanent wilting point and depends on the soil type. Therefore, $W_{wilt}$ is in in each grid point identical in all three simulations. $W_{root}$ is the water content within the rooted soil depth. This quantity is different at each grid point and changes at each time step of the simulation. Thus, it is from our point of view not meaningful to include these quantities in table 2.

**Papers:**

Rohatyn, S., et al. (2018). "Differential Impacts of Land Use and Precipitationon "Ecosystem Water Yield"." Water resources research 54(8): 5457-5470.

Baldocchi Dennis, Qi Chen, Xingyuan Chen, Siyan Ma, Gretchen Miller, Youngryel Ryu, Jingfeng Xiao, Rebecca Wenk and John Battles (2009). "The Dynamics of Energy, Water and Carbon Fluxes in a Blue Oak (Quercus douglasii) Savanna in California,USA", in: "Ecosystem Function in Global Savannas: Measurement and Modeling at Landscape to Global Scales" – edited by Michael J. Hill and Niall P. Hanan, CRC/Taylor and Francis.

Ryu Youngryel, Dennis D. Baldocchi, Siyan Ma and Ted Heh (2008), "Interannual variability of evapotranspiration and energy exchange over an annual grassland in California", JOURNAL OF GEOPHYSICAL RESEARCH, VOL. 113.

---

## Author Comment (AC2) · 23 Oct 2020

The authors have identified an important, and poorly understood, aspect of the effects of afforestation/deforestation in temperate latitudes on climate: do forests increase or decrease evapotranspiration (ET) compared with grasslands? Some observational studies suggest forests have greater rates of ET; some show the opposite. Many modeling studies show forests increase ET; others do not. The topic is fraught with confusion. I had hoped this manuscript would clarify the science and provide a strong, insightful understanding of forest ET and the factors controlling ET. However, by using a poorly documented model, and by not adequately describing the model, the rationale for parameterizations and parameter values, and the limitations of the model, the manuscript does not clarify the science and, instead, adds more confusion to the literature.
- thank you very much for your detailed comments. We hope that our responses will help to dissolve potential confusions.

1. I have several concerns about the VEG3D model. From the few equations given in the manuscript, it appears to be a highly simplified land surface model. There is nothing wrong with that! But I suspect the findings of the study do not extend to more complex land surface models. The authors need to provide a thorough description of the model, justify the parameterizations used in the model, and justify parameter values. They also need to discuss how the simplifications of VEG3D might limit the generality of the results. This is not the first time this issue has arisen. VEG3D was used in a previous study by the lead author:

Breil, M., and Coauthors, 2020: The Opposing Effects of Reforestation and Afforestation on the Diurnal Temperature Cycle at the Surface and in the Lowest Atmospheric Model Level in the European Summer. J. Climate, 33, 9159–9179

In full disclosure, I was a reviewer of that manuscript and noted in my review thatVEG3D is a poorly documented model, is not widely known by the scientific community, and has not been tested in temperate forest/grassland simulations in comparison with flux tower measurements. That does not mean that the model is deficient or in-appropriate for this study, but the description of ET provided in the current manuscript reveals some non-standard formulations in the model that likely limit the generality of the results.
- in the revised version of the manuscript, the applied parameterizations and parameter values are discussed in more detail (see answers to the following comments).

1a. The authors describe the aerodynamic resistance ra used in the transpiration equation (eq 3). This is not a standard formulation of aerodynamic resistance (I have never seen it before). The resistance depends on wind speed at the top of the canopy, leaf area index, and some undescribed parameters. Classic textbooks on micrometeorology and boundary layer meteorology formulate the resistance using integrated flux-profile relationships between the apparent source/sink in the canopy (at a height equal to the roughness length [z0] plus displacement height [d]) and the lowest model level in the atmosphere

$$[z]:ra = [\ln(z-d)/z0]^2 / (k^2 * u)$$

u is wind speed at z. Depending on the specific model, z0 can be either that for momentum or for scalars, and ra is adjusted for atmospheric stability. What is the justification for eq 3, which seems to go back to two very old papers (Deardorff, 1978; Taconet et al., 1986)? Why is this equation used

rather than classic boundary layer theory? It seems from eq 3 that roughness length only enters the model through wind speed at the top of the canopy (uaf), but there is no equation for uaf. It appears to go back to Goudriann's old work. This is very important, because the key outcome of the study is that surface roughness is the primary difference between forests and grasslands. Readers must understand precisely how surface roughness is used in the model and why particular formulations are used in the model.

- we are sorry that Eq. (3) caused that much confusion. This is exactly the opposite of what we intended, by using a simple description of $r_a$. We thought that this formulation would keep the section clear and understandable. Obviously, this was not the case. The description of $r_a$ is therefore refined as follows:

Lines (86-105):
"In $r_a$, the turbulent atmospheric conditions for the transfer of water vapor are reflected, which are calculated by means of an empirical parameter $C_{leaf}$ and the friction velocity $u*$:

$$r_a = \frac{C_{leaf}}{u*}$$
Eq. (3)

$C_{leaf}$ describes an empirical interrelation between the turbulent exchange and the Leaf Area Index (LAI) (Taconet et al., 1986), in relation to the leaf geometry, represented by the plant specific parameter $c_{veg}$ (a) (Goudriaan, 1977):

$$C_{leaf} = \frac{1 + 0.5 * LAI}{0.04 * LAI * c_{veg}}$$
Eq. (4)

$u*$ is classically derived from the Monin-Obukhov Similarity Theory (Monin and Obukhov, 1954) and as such mainly dependent on $z_0$:

$$u* = k \frac{\left(v_{z_a} - v_{z_0}\right)}{\ln\left(\frac{z_a}{z_0}\right) + \Psi\left(\frac{z_a}{L*}\right) + \Psi\left(\frac{z_0}{L*}\right)}$$
Eq. (5)

where $z_a$ is the height of the lowest atmospheric model level and $z_0$ is the roughness length. $v_{za}$ and $v_{z0}$ are consequently the wind velocities at the respective heights. $k$ is the Karman-constant. $L*$ is the Monin-Obukhov length and $\Psi$ is a stability-function according to Businger et al., (1971), establishing empirical relationships in turbulent motion, which depend on the atmospheric stratification. According to Goudriaan (1977), $r_a$ and consequently its contribution to the transfer coefficient c, is primarily influenced by one vegetation parameter: the surface roughness (b)."

In comparison to the previous version, it is shown that $r_a$ depends on $u*$ and the empirical parameter $C_{leaf}$, representing an empirical interrelation between the turbulent exchange and the Leaf Area Index (LAI), in relation to the leaf geometry. The calculation of $r_a$ is therefore totally in line with the classic boundary layer theory.

In connection with the empirical $C_{leaf}$ parameter, a plant specific parameter $c_{veg}$ is now introduced, representing the leaf geometry. This parameter was not mentioned in the previous version of the manuscript, since its impact on $r_a$ is small in comparison to the surface roughness (Goudriaan, 1977). But due to the refinement of this section, this parameter is now additionally introduced and discussed in the course of the manuscript.

Admittedly, the description of the empirical vegetation parameter $C_{leaf}$ is not the latest one. But this does not mean that the produced results are not valid. On the contrary; within the scope of several

model-intercomparison studies, it could be demonstrated that VEG3D produces comparable results to more recent LSMs (e.g. Davin et al., 2020; Breil et al., 2020; Krinner et al, 2018).

1b. The formulation of canopy resistance to transpiration (eq 4) is also somewhat odd. It goes back to an equation in Deardoff (1978), in which canopy resistance depends on a specified minimum resistance that is modified for solar radiation and soil moisture. Most current-generation land surface models use an approach that couples photosynthesis and stomatal conductance through the Farquhar et al. (1980) photosynthesis model and semi-empirical stomatal conductance models such as proposed by Ball-Berry or Medlyn. In addition to light and soil moisture effects on stomatal conductance, those models also include temperature and vapor pressure deficit (VPD) effects on stomatal conductance. The VEG3D model ignore those latter two effects. That exclusion greatly limits the generality of the main finding of the study (that VPD, as modified by surface roughness, is a key determinant of differences in ET between forests and grasslands). The response of stomata to VPD is not considered (i.e. stomata close as VPD increases). Nor are the indirect effects of VPD on stomata through leaf temperature considered. Again, readers need to know why eq 4 is used in contrast with more common stomatal conductance models and what the implications of eq 4 are for the main findings of the study.
- You are right, temperature and vapor pressure deficit effects on stomatal conductance are not considered in VEG3D. These particular capabilities of trees can certainly affect evapotranspiration rates in regions with pronounced differences in the saturation deficit between forests and grasslands, like Southern Europe.
Interestingly, the results of model-intercomparison studies show that LSMs, in which these stomatal effects are integrated, exhibit comparable evapotranspiration responses as VEG3D (e.g. Davin et al., 2020). For instance, in the framework of the LUCAS, simulations with the classic model VEG3D and the more sophisticated Community Land Model under the same atmospheric boundary conditions, show similar spatial patterns of increased or reduced evapotranspiration rates with afforestation (Davin et al., 2020).Thus, the differences in the model complexity (effects of shaded and unshaded leaves or the vapor pressure dependency of stomata closure) cannot be the main reason for the simulated differences in evapotranspiration responses of forests and grasslands. These different evapotranspiration responses must rather be caused by a fundamental mechanism, which is simulated in both, classic as well as complex LSMs. This is now emphasized in the manuscript.

Lines (34-37):
"According to our present knowledge about the biogeophysical effects of forests and grasslands, this increased forest evapotranspiration is caused by deeper roots (Schenk and Jackson, 2003) and a higher Leaf Area Index (LAI, e.g. Henderson-Sellers, 1993) than in grassland, whose influence can be attenuated by a reduced photosynthetic activity of forests and an associated stomata closure (Leuzinger et al., 2005)."

Lines (351-357):
"Another possible reason for the disagreement between the simulation results and the observations is the missing consideration of vapor pressure effects on the stomatal resistance in CCLM-VEG3D. For instance, in Southern Europe the saturation deficit of forests is particularly lower than for grasslands. In contrast to the simulated trees in CCLM-VEG3D, real trees are potentially able to adapt to this lower saturation deficit, by further reducing the stomatal closure and thus the transfer coefficient. In line with the introduced evapotranspiration concept, the transpiration facilitating characteristics of forests (2) would be further enhanced, counteracting the reduced saturation deficit (1) in Southern Europe and thus, would increase forest evapotranspiration."

Lines (314-328):
"Climate simulations with incorporated Land Surface Models (LSMs) are an appropriate method to analyze the reasons for these varying evapotranspiration rates of forests and grasslands. However,

models constitute only a simplified description of reality and thus, cannot represent the complex biogeophysical processes in nature comprehensively. For instance, VEG3D does not consider the effects of the multilayer canopy structure of trees (effects of shaded and unshaded leaves; Bonan et al., 2012) or the influence of the understory on evapotranspiration rates, which can contribute substantially to total evapotranspiration in forests (e.g. Yepez et al., 2003). Furthermore, VEG3D does not consider the impact of temperature and vapor pressure deficit on stomata closure, potentially affecting evapotranspiration rates in regions with pronounced differences in saturation deficit between forests and grasslands. But the results of model-intercomparison studies show that more sophisticated LSMs, in which these biogeophysical effects are integrated, exhibit comparable evapotranspiration responses to afforestation as VEG3D (e.g. de Noblet-Ducoudré et al. 2012; Davin et al., 2020). For instance, in the framework of the LUCAS project, simulations with the classic model VEG3D and the more sophisticated Community Land Model under the same atmospheric boundary conditions, show similar spatial patterns of increased or reduced evapotranspiration rates with afforestation (Davin et al., 2020). Thus, the differences in the model complexity (effects of shaded and unshaded leaves or the vapor pressure dependency of stomata closure) cannot be the main reason for the simulated differences in evapotranspiration responses of forests and grasslands. These different evapotranspiration responses must rather be caused by a fundamental mechanism, which is simulated in both, classic as well as complex LSMs."

In order to get to the bottom of these fundamental processes, the use of a less complex model can even be beneficial. In such a model, the degrees of freedom are reduced and functional interrelations can consequently be deduced more easily. For this reason, we are able to show in the manuscript that the driving force behind evapotranspiration (saturation deficit) is already reduced in forests (in comparison to grasslands), due to their inherent biogeophysical characteristics ($z_0$). Depending on latitude and forest type, therefore, forests can have lower evapotranspiration rates than grasslands.

1c. The term (1+0.5*LAI)/LAI is common to both ra and rc. What does this term represent? It seems to be a scaling term for canopy LAI (i.e, from a leaf resistance to a canopy resistance). Aerodynamic resistance is commonly expressed per unit ground area. Why does ra need to be scaled by LAI?
- This term represents an empirical interrelation between the turbulent exchange and the vegetation specific characteristics *LAI* and the leaf geometry (Taconet et al., 1986). The term accounts for the fact that the turbulent exchange is proportional to the LAI with a low exchange for small LAIs and an increasing exchange with increasing LAI (but the interrelation has an upper limit; 0.5).

2. The authors emphasize that differences between forest and grassland arise in terms of five model parameters: surface roughness, albedo, root depth, leaf area index, and minimum stomatal conductance.

2a. The justification for several parameter choices goes back to papers by Garratt (1993) and Henderson-Sellers (1993). There has been a lot of model development since then. How do these parameter choices compare with values used in the current generation of land surface models?
- the used model parameters in VEG3D are very similar to the parameters used in other Land Surface Models as it is shown in Breil et al., (2020) (see table below). The albedo and $z_0$ values are totally in line with the values in other models. The LAI values in VEG3D are higher than in the other models. But the relative LAI differences between the different land use classes (coniferous forest, deciduous forest, grassland) are again comparable. For instance, the relative difference in the contribution of the LAI to $r_a$ (caclulated via Eq. (4)) between coniferous and deciduous forest is 0.97 in VEG3D. If one would use instead the LAI values used in the Community Land Model (CLM), the relative difference would be 0.96. A similar picture is drawn for the relative differences between coniferous forests and grasslands. In VEG3D, the relation is 0.81, while the use of the

CLM values would result in a relation of 0.83. Thus, it can be stated that the parameter values in the respective LSMs lead to comparable physical dependencies.

TABLE 1. Surface roughness $z_0$, leaf area index (LAI), and surface albedo $\alpha$ in summer (yearly maximum) used in each LUCAS-Ensemble member for needleleaf evergreen trees (NET), broadleaf deciduous trees (BDT), and C3-type grassland (C3).

| | $z_0$ | | | LAI | | | $\alpha$ | | |
|---|---|---|---|---|---|---|---|---|---|
| | NET | BDT | C3 | NET | BDT | C3 | NET | BDT | C3 |
| WRF-NoahMP | 1.09 | 0.8 | 0.12 | 4 | 4.7 | 3.5 | 0.11[a] | 0.13[a] | 0.23[a] |
| WRF-CLM4.0 | 0.7 | 0.83 | 0.048 | 3.75 | 3.38 | 2.38 | 0.11[a] | 0.13[a] | 0.21[a] |
| CCLM-VEG3D | 1 | 0.8 | 0.03 | 9 | 8 | 4 | 0.11 | 0.15 | 0.2 |
| CCLM-TERRA | 1 | 1 | 0.03 | 8 | 6 | 4.5 | 0.1 | 0.15 | 0.2 |
| CCLM-CLM4.5 | 0.7 | 0.83 | 0.048 | 3.75 | 3.38 | 2.38 | 0.11[a] | 0.13[a] | 0.21[a] |
| REMO-iMOVE | 1.4 | 1 | 0.05 | 5 | 5 | 3 | 0.155 | 0.175 | 0.21 |

[a] Calculated for an exemplary leaf/stem ratio.

2b. Table 2 shows only a small difference in rmin between forest and grassland, and no difference between coniferous and deciduous forest. What is the justification for the parameter values? Are there physiological measurements that support them? The values for rmin are very important to the results of the study. The relative contributions of aerodynamic resistance and canopy resistance to total resistance determine the model sensitivity to roughness length. The fact that rmin is similar for all vegetation precludes physiological differences in stomatal conductance from determining differences in ET.

- the parameter values used in VEG3D are based on the results of several studies (see figure below from the review paper of Garratt (1993)). The range of rmin values is, in this context, quite large for the different land use types. Therefore, in VEG3D an average rmin value is used for each vegetation type. In general, rmin values of forests are smaller than rmin values of grasslands. rmin values of coniferous forests are on the same level as rmin values of deciduous forests. The generally lower rmin values of forests in comparison to grasslands are an important point in the study. Due to this, rmin of forests also facilitates transpiration as the other vegetation specific characteristics do in VEG3D and thus counteracts the reduced vapor pressure deficit.

[Figure]

FIG. 4. Variation of unconstrained stomatal resistance $r_s^*$ and minimum surface resistance $r_s$ (min) with vegetation type, based on observations, literature values and model-specified values. Symbols are as follows: (a) (○) from Dorman and Sellers (1989); (●) from Shuttleworth et al. (1984b) for 3 forest sites; (×) from Sellers and Dorman (1987) for barley; (N) from Noilhan and Planton (1989) for maize (showing a wide range in resistance), oats (anomalously high) and a pine forest; 1) barley, 2) wheat, 3) maize, 4) spruce forest are all from Sellers and Dorman (1987). (b) (○) from Dorman and Sellers (1989); (●) from Shuttleworth et al. (1984b); (×) from Sellers and Dorman (1987) for maize; (△) from Sellers and Dorman (1987) for spruce; (G) from Garratt (1978) for subtropical savannah; (P) from Perrier (1982).

2c. No details are given on how root depth affects transpiration, or how the root depth parameter is used in the model. The root depth of deciduous forest is twice that of coniferous forest. Is this the reason for the differences between deciduous and coniferous forests when they are converted to grassland?

- Thanks for raising this important aspect. As described in section 3.2, differences between coniferous and deciduous forests are caused by the lower c value in deciduous forests in comparison to coniferous forests (statement with respect to Central Europe, where opposing evapotranspiration responses occur between coniferous and deciduous forest for the same latitude and vapor pressure deficit).

Both forests types have higher c values than grassland. Since the vapor pressure deficit is in Central Europe for both forest types smaller than for grassland (and thus the driving force for evapotranspiration), the c value of coniferous forest must be higher than the one of deciduous forest, leading to higher ET in coniferous forests and lower ET in deciduous forests in comparison to grasslands. Deeper roots (as for deciduous forests), lead in times of a reduced water availability to increased c values. So if the root depths would be the reason for the different responses, deciduous forests should have higher ET than coniferous forests. Thus, the impact of the root depths on the c calculation must be smaller than the impact of e.g. the albedo and the LAI (albedo is higher and LAI is lower in deciduous forest and the c values consequently smaller). This discussion is now included in the manuscript.

Lines (244-258):

"In Central Europe, the saturation deficit in the FOREST run is comparable to Northern Europe. But in contrast to Northern Europe, regions of increased evapotranspiration are simulated as well as regions of reduced evapotranspiration compared to the GRASS simulation (Figure 2a). As already mentioned in section 3.1, the regions of increased evapotranspiration coincide with regions covered by coniferous forests, while regions of reduced evapotranspiration are covered by deciduous forests. Since the saturation deficit reduction in the FOREST run is comparable for both forest types in Central Europe (Figure 4a), these different evapotranspiration responses to afforestation must be associated with differences in the transfer coefficient $c$ (Eq. 1). The transfer coefficient $c$ of coniferous forest must therefore be higher than the one of deciduous forest. In a coniferous forest $LAI$ is increased and albedo is reduced in comparison to a deciduous forest, while in deciduous forest the root depth and $c_{veg}$ are increased. Thus, both forest types have characteristics which lead to high $c$ values. However, since evapotranspiration in Central Europe is higher for coniferous forests than for deciduous forests, the impact of $LAI$ and the albedo (pronounced in coniferous forests) on $c$ must be higher than the impact of the root depth and $c_{veg}$ (pronounced in deciduous forests). As a result, the impact of the higher transfer coefficient $c$ of coniferous forests surpasses the effects of the lower saturation deficit in Central Europe in the transpiration flux calculation and evapotranspiration is increased, while for deciduous forests the impact of the reduced saturation deficit is dominating and evapotranspiration is reduced."

We are absolutely aware that these parameter values are associated with uncertainties and that the use of only two different forest types is simplified. Therefore, we do not intend to assess the transpiration rates of individual forest types. The aim of this study is to explain the different evapotranspiration rates of forests and grasslands in a physically consistent way. In this context we can show that the evapotranspiration response generally depends on the forest characteristics (without assessing specific forest types).

2d. No details are given for albedo. What is the radiative transfer parameterization in the model? Land surface models typically simulate radiative transfer for visible and near-infrared wavebands and for direct and diffuse radiation. Albedo is a complex result of leaf and stem reflectances, leaf and stem area index, solar zenith angle, and soil moisture. Because only a single albedo is listed as a parameter in Table 2, this makes me think there is no such complex radiative transfer

parameterization in VEG3D and instead the model uses a bulk surface albedo that is prescribed as a parameter. Readers need further information.

-You are right, in VEG3D a bulk approach is used for the albedo. This is clarified in the revised manuscript (Lines 115-116).

3. A striking aspect of Figure 2 is the difference between coniferous and deciduous forests when replaced with grassland. Summer latent heat fluxes are larger in coniferous forest compared with grassland but are smaller than grassland in deciduous forest. This pattern is universally consistent throughout the domain, except for southern Spain and Turkey (smaller latent heat fluxes compared with grassland in a mostly coniferous forest region). The authors acknowledge the influence of forest type (lines165-166), but for the most part discuss their results in terms of Northern Europe versus Southern/Central Europe. For example, the authors frame their conclusions as: "In Northern Europe evapotranspiration is increased with afforestation, in Southern and Central Europe evapotranspiration is decreased" (lines 261-262). The differing results of coniferous and deciduous forests are not even mentioned in the abstract. I would like to see more of a discussion of coniferous versus deciduous forests.

- thanks for this suggestion. A central message of this study is that the interplay between factor 1 (vapor pressure deficit) and factor 2 (high c values due to transpiration facilitating characteristics of forests) is controlled by two determinants. These are the latitude (Lines 337-348) and the forest type (Lines 358-369). This is also stated in the last sentence of the abstract (Line 27). Apparently, we were not able to express this clearly in the manuscript. Therefore, we add some discussion to our original statements.

Lines (244-258):

"In Central Europe, the saturation deficit in the FOREST run is comparable to Northern Europe. But in contrast to Northern Europe, regions of increased evapotranspiration are simulated as well as regions of reduced evapotranspiration compared to the GRASS simulation (Figure 2a). As already mentioned in section 3.1, the regions of increased evapotranspiration coincide with regions covered by coniferous forests, while regions of reduced evapotranspiration are covered by deciduous forests. Since the saturation deficit reduction in the FOREST run is comparable for both forest types in Central Europe (Figure 4a), these different evapotranspiration responses to afforestation must be associated with differences in the transfer coefficient $c$ (Eq. 1). The transfer coefficient $c$ of coniferous forest must therefore be higher than the one of deciduous forest. In a coniferous forest *LAI* is increased and albedo is reduced in comparison to a deciduous forest, while in deciduous forest the root depth and $c_{veg}$ are increased. Thus, both forest types have characteristics which lead to high $c$ values. However, since evapotranspiration in Central Europe is higher for coniferous forests than for deciduous forests, the impact of *LAI* and the albedo (pronounced in coniferous forests) on $c$ must be higher than the impact of the root depth and $c_{veg}$ (pronounced in deciduous forests). As a result, the impact of the higher transfer coefficient $c$ of coniferous forests surpasses the effects of the lower saturation deficit in Central Europe in the transpiration flux calculation and evapotranspiration is increased, while for deciduous forests the impact of the reduced saturation deficit is dominating and evapotranspiration is reduced."

Lines (358-369):

"On the other hand, the simulation results show that the balance between factor (1) and (2) is differently pronounced for different forest types. In Central Europe, for instance, deciduous and coniferous forests are showing opposing evapotranspiration responses to afforestation, although they are facing a comparable saturation deficit (1). Differences in the evapotranspiration rates must consequently be associated with differences in the transfer coefficents (2). A deciduous forest, for instance, has a lower LAI and higher albedo values than a coniferous forest (Table 2). The transfer coefficient is consequently lower and factor (2) is becoming weaker. The impact of the saturation deficit (1) is therefore dominating the effects of factor (2) and the transpiration rates of deciduous

forests are reduced compared to grassland in Central Europe. But for coniferous forest, which are facing a similar saturation deficit (1), the impact of factor is increased (2), due to their higher LAI and lower albedo values. The transpiration rates are consequently higher for coniferous forests in this region. These results are also in line with observation-based studies, showing that evapotranspiration rates differ between different forest types (e.g. Brown et al., 2005), whereby higher evapotranspiration rates are generally assigned to coniferous forests (e.g. Teuling, 2018). Furthermore, Marc and Robinson, (2007) showed that also the age of the forest affects evapotranspiration."

As already mentioned in our response to comment 2c, we do not intend to give specific statements about the evapotranspiration rates of individual forest types, due to the uncertainties related to the used parameter values.

3a. What, specifically, are the differences between these forests that cause the results? One generally thinks of coniferous forests as having a more conservative water-use strategy than deciduous forests (seen, for example, in higher stomatal resistance). But both forests have the same minimum resistance. Is the different response related to rooting depth? In their analysis of soil water (Figure 3), the authors suggest it is not but the analysis is not definitive. It would be better to look at the soil moisture stress term in canopy resistance.
- as already mentioned in comment 2c, the different behavior of coniferous and deciduous forests in Central Europe is mainly caused by the higher albedo values and lower LAI values.
In Figure 3, two important features of the FOREST and GRASS simulations can be seen; (1) the contribution of soil evaporation to total evapotranspiration is the same for forests and grasslands and differences in total evapotranspiration must consequently be caused by transpiration, (2) the available amount of soil water for evapotranspiration is higher for forests than for grasslands. Figure 3a shows that this water amount is lower in coniferous forests than in deciduous forests in Central Europe, due to the more shallow roots. As already discussed in comment 2c, this should lead in deciduous forests to a higher c value compared to coniferous forests, if all other vegetation characteristics would be the same. But since albedo is higher and LAI is lower, this effect of the root depths on the c value is compensated.

3b. Are the results consistent with observations? What do flux towers show? What does MODIS ET show (but remember that MODIS ET is a modeled product).
- as discussed in section 4, the model results have comparable features to observations (dependency to the latitude (Li et al., (2015), to the forest type (Teuling, 2018) and the increased evapotranspiration rates in Northern Europe (Duveiller et al., 2018), Lines 345-347; Line 366-369). The validity of a direct and detailed comparison with paired measurement sites is from our point of view limited. In contrast to the local impacts of land use on the evapotranspiration rates as reflected in observations, in our simulations large-scale forestation scenarios are applied reflecting the general and idealized evapotranspiration responses to forestation. This is also true for the comparison with satellite-based data (e.g. MODIS), although general features of such data (e.g. latitude dependency) are reproduced.

Lines (379-384)
"However, a direct comparison of the CCLM-VEG3D model results with observational data is generally difficult, due to the different spatial representativity of the data. While observational data (satellite data as well as data from eddy covariance flux towers) reflect the local transpiration responses to forestation (Bright et al., 2017), in the CCLM-VEG3D simulation setup, large-scale forestation scenarios are applied to analyze the general transpiration responses to forestation in an idealized and isolated way. Therefore, it is difficult to assess the CCLM-VEG3D model results quantitatively and qualitatively in comparison to observations"

Therefore, the aim of our study is rather to introduce a concept of the physical reasons for the deviating evapotranspiration responses of forests and grasslands, than reproducing observed evapotranspiration rates of coniferous and deciduous forests in specific regions.

4. The crux of the study is Figure 4, which shows the difference in saturation deficit between the forest and grassland simulations. Saturation deficit decreases for forests throughout Europe, with a particularly large decrease in latitudes south of about 40N. The authors discuss the results in light of VPD, resistances, and other parameters that affect transpiration (lines 219-234). No data or figures are provided to justify the interpretation. Skeptical readers need to see more evidence that supports the argument if they are to believe the study.

- As stated in Eq. 1, evapotranspiration is controlled by two factors, (1) the vapor pressure deficit and (2) the transfer coefficient c. Since forests have all over Europe a higher c value than grasslands (table 1 and table 2), lower evapotranspiration rates can only be explained by a lower vapor pressure deficit. Regions of higher or lower evapotranspiration rates than grasslands must therefore inevitably be caused by the interplay of both factors.

5. Figure 5d: Why does net shortwave radiation change when roughness length is changed to that of grassland?

- The albedo is identical in the FOREST and the ROUGH simulation. Therefore, differences in the net short-wave radiation must be caused by atmospheric processes. In Northern Europe, the cloud cover is increased in FOREST (Figure 5c in the revised manuscript), due to the higher evapotranspiration rates in comparison to ROUGH and thus, net short-wave radiation is reduced (Figure 5d). In consequence, the temperature reduction in FOREST is comparatively strong pronounced (Figure 6c) and the increase in evapotranspiration is attenuated (Figure 2b). In the revised paper, this atmospheric feedback process is included.

Lines (266-272):
"Differences in evapotranspiration as seen for the FOREST and GRASS runs (Figure 2), inevitably affect the atmospheric conditions in these simulations. For instance, the increased evapotranspiration rates in Northern Europe in FOREST lead to an increased cloud cover in this region (Figure 5a). The incoming solar radiation is consequently reduced in comparison to GRASS. However, since the albedo of the trees in the FOREST simulation is lower than the albedo of grassland in the GRASS run, the reduction of the incoming solar radiation is compensated and net short-wave radiation is slightly increased in Northern Europe (Figure 5b). For the rest of the European continent, this albedo effect is even stronger pronounced and the net short-wave radiation is considerably increased, since cloud cover is not changed compared to GRASS."

Lines (282-288):
"Due to the increased evapotranspiration rates in ROUGH in Northern Europe (Figure 2b), cloud cover is increased in this region in comparison to the FOREST run (Figure 5c). The net short-wave radiation is consequently slightly reduced (Figure 5d). But for the rest of the European continent, net short-wave radiation in FOREST and ROUGH is on the same high level, due to the unchanged albedo values. The reduced surface roughness in ROUGH reduces all over Europe the sensible heat transport into the atmosphere (Figure 6d). Thus, the high radiative energy is not as efficiently transformed and transported into the atmosphere as in FOREST, with the consequence that the surface temperatures are increased, similarly to the GRASS simulation (Figure 6c)."

6. Lines 288-290: The authors state that "the dependency of the evapotranspiration rates of forests and grasslands on the latitude is also documented in satellite observations (e.g. Li et al., 2015), showing for example higher evapotranspiration rates of grasslands in South-Eastern Europe, while in Central and Northern Europe evapotranspiration is lower than in forests (Duveiller et al., 2018)". Li et al used MODIS ET, which is a modeled product. What did Duveiller et al base their analysis

on? And, remember, that the more striking aspect of Figure 2 is not the latitudinal dependence but the difference between coniferous and deciduous forests. What do observations say about that difference?

- Duveiller et al., (2018) use also MODIS data, which is as you already mentioned not completely observation-driven. Regarding the evapotranspiration differences between coniferous and deciduous forests, therefore, we think that it is the best to rely in this case on direct measurements, such as lysimeter data. Evapotranspiration rates from these data sets are quite in contradiction to satellite/model products. For instance, while satellite/model products assign higher evapotranspiration rates to deciduous forests (Duveiller et al., 2018), direct measurements at lysimeter stations assign higher evapotranspiration rates to coniferous forests (Teuling, 2018).

Lines (366-369):
"These results are also in line with observation-based studies, showing that evapotranspiration rates differ between different forest types (e.g. Brown et al., 2005), whereby higher evapotranspiration rates are generally assigned to coniferous forests (e.g. Teuling, 2018)."

Additional References:

Businger, J. A., Wyngaard, J. C., Izumi, Y., and Bradley, E. F., (1971): Flux-Profile Relationships in the Atmospheric Surface Layer, *J. Atmos. Sci.*, 28, 181–189.

Krinner, G., and Coauthors, (2018): ESM-SnowMIP: assessing snow models and quantifying snow-related climate feedbacks. *Geoscientific Model Development, 11*, 5027-5049.

---

## Author Response (AR2)

The authors have done an excellent job at responding to my comments. The VEG3D model is much clearer now. I have a few minor comments that would further strengthen the manuscript, but I leave it up to the authors to decide whether or not to include these revisions.
- Thank you very much for your positive assessment and constructive suggestions.

1. I am a fan of simplified models. In their response to comment 1b, the authors provide a strong defense for simplified models with the statements: "In order to get to the bottom of these fundamental processes, the use of a less complex model can even be beneficial. In such a model, the degrees of freedom are reduced and functional interrelations can consequently be deduced more easily." This is never said in the manuscript. What not acknowledge up front in that VEG3D is a simplified model? This could be added in lines 62-69. Or perhaps the authors do not think VEG3D is simplified?
- thanks for this suggestion. We added the following statement about the benefits of less complex models to the manuscript.

Lines (329-334):
"These different evapotranspiration responses must rather be caused by a fundamental mechanism, which is simulated in both, classic as well as complex LSMs. In order to get to the bottom of these fundamental processes, the use of a less complex model can even be beneficial. In such a model, the degrees of freedom are reduced and functional interrelations can consequently be deduced more easily. Therefore, by means of a sensitivity study with this less complex CCLM-VEG3D model, in which the surface roughness of forests was reduced to grassland (ROUGH), this fundamental mechanisms behind the varying evapotranspiration rates of forests and grasslands could be clearly revealed:"

2. In their response to comment 2a, the authors reference Table 1 of Breil et al. (2020) to justify parameter choices (comparable to those used in other models). Why not say this in the manuscript, too?
- we added the following statement to the revised manuscript:

Lines (123-124):
"The values of these six vegetation parameters in VEG3D are in line with the parameter values used in other state of the art LSMs (Breil et at., 2020)."

3. The revised description of the aerodynamic resistance is good. One minor suggestion is to change "reflected" to "included". A quick reading of the sentence might make readers think that the "transport of water vapor" is a reflection from the surface. What you are really saying is that ra accounts for turbulent transport of water vapor. Also, it is not clear what (a) refers to following cveg (line 91) and (b) following surface roughness (line 105). My reading of the equations is that three vegetation parameters affect ra: LAI, cveg, and z0, but that (citing Goudriaan 1977) z0 is thought to be the most important. If this is the intent, it could be said more clearly.
- we changed "reflected" to "included", according to your suggestion. Furthermore, we agree that this paragraph could be written more clearly. This is done in the following way.

Lines (104-106):
"According to Eq. (4) and Eq. (5), ra is consequently affected by three vegetation parameters, namely a plant specific parameter cveg (a), the surface roughness z0 (b) and the LAI (c). But out of these three parameters, the influence of the surface roughness (b) on ra and thus, on the transfer coefficient c, is clearly dominating (Goudriaan 1977)."

**Anonymous Referee #1**

I would like to thanks the Authors for replying and resending the paper.
- Thank you very much for your positive assessment and constructive suggestions.

However, at this stage I do have one main comments, which I found needed to the emerging results: This is with regard to the comment on the differences between observation and model outcomes in the large dry region. In their answer the authors commented that "…While observational data reflect the local differences between forest and grassland transpiration rates, in our simulation setup, large-scale forestation scenarios are applied to analyze the general transpiration responses to forestation in an idealized and isolated way. It is therefore very difficult to assess the model results quantitatively and qualitatively…".

Then that the: "… eddy covariance flux towers) reflect the local transpiration responses to forestation (Bright et al., 2017), in the CCLM-VEG3D simulation setup, large-scale forestation scenarios are applied to analyze the general transpiration responses to forestation in an idealized and isolated way …".

And last (which was not much of clear for me) that: "…a physically consistent explanation for this phenomena, in which the evapotranspiration responses are described as an interplay of two factors, namely the reduced vapor pressure deficit in forests facing their evapotranspiration facilitating biogeophysical characteristics. …"

These statements need explanations to what behind the differences and best, if exist, to provide evidences for screen (i.e., air a 2 m) not surface (i.e., skin; Fig. 6) temperature and humidity differences between forest to grass ecosystems in published papers.
- it seems that we were not able to satisfactorily demonstrate that model results and observational data are difficult to compare, regarding the effects of land use changes. Therefore, the corresponding section is rephrased and the included statements are underpinned by additional references.

Lines (383-388):
"However, it is generally difficult to assess the effects of afforestation by a direct comparison of the CCLM-VEG3D model results with observational data, due to discrepancies on the scale of processes considered in models and observations (Davin et al., 2020). In observational data (satellite data as well as data from eddy covariance flux towers) forests and grasslands in immediate vicinity are compared. Differences in the measured fluxes are therefore directly related to the local land cover differences (Bright et al., 2017). In contrast, differences in model results for forests and grasslands are additionally affected by large-scale atmospheric feedback processes (Winckler et al., 2017)."

Since part of the differences between the ecosystems are not included in the model I would even recommend to consider limiting the model to the seasons within the model capability to assess more directly the vegetation performance.
- the study is already focusing on the summer season

The sentences in lines 374-8 are unclear
- this paragraph is rephrased:

Lines (377-382):
"Since this weighting is model-specific, slightly different evapotranspiration responses of forests and grasslands are anticipated for different model simulations. Moreover, different evapotranspiration responses can also be expected within observational data, since the biogeophysical characteristics of forests and grasslands vary also in nature (Garratt, 1993; Henderson-Sellers, 1993; Schenk and Jackson, 2003). Taking these uncertainties into account

differences between the CCLM-VEG3D results and observations, as present in Southern Europe (Rohatyn et al., 2018), can potentially be explained."